# Evaluation of 6-Hydroxydopamine and Rotenone In Vitro Neurotoxicity on Differentiated SH-SY5Y Cells Using Applied Computational Statistics

**DOI:** 10.3390/ijms23063009

**Published:** 2022-03-10

**Authors:** Rui F. Simões, Paulo J. Oliveira, Teresa Cunha-Oliveira, Francisco B. Pereira

**Affiliations:** 1CNC-Center for Neuroscience and Cell Biology, CIBB-Centre for Innovative Biomedicine and Biotechnology, University of Coimbra, 3004-504 Coimbra, Portugal; rui.simoes@uc-biotech.pt (R.F.S.); pauloliv@cnc.uc.pt (P.J.O.); 2Programme in Experimental Biology and Biomedicine (PDBEB), Center for Neuroscience and Cell Biology, Institute for Interdisciplinary Research, 3004-504 Coimbra, Portugal; 3CISUC, Centre for Informatics & Systems, University of Coimbra, 3030-299 Coimbra, Portugal; xico@dei.uc.pt; 4Polytechnic Institute of Coimbra, Coimbra Institute of Engineering, 3030-190 Coimbra, Portugal

**Keywords:** 6-hydroxydopamine, rotenone, in vitro neurotoxicity, mitochondrial dysfunction, exploratory data analysis, applied computational statistics, unsupervised and supervised machine learning

## Abstract

With the increase in life expectancy and consequent aging of the world’s population, the prevalence of many neurodegenerative diseases is increasing, without concomitant improvement in diagnostics and therapeutics. These diseases share neuropathological hallmarks, including mitochondrial dysfunction. In fact, as mitochondrial alterations appear prior to neuronal cell death at an early phase of a disease’s onset, the study and modulation of mitochondrial alterations have emerged as promising strategies to predict and prevent neurotoxicity and neuronal cell death before the onset of cell viability alterations. In this work, differentiated SH-SY5Y cells were treated with the mitochondrial-targeted neurotoxicants 6-hydroxydopamine and rotenone. These compounds were used at different concentrations and for different time points to understand the similarities and differences in their mechanisms of action. To accomplish this, data on mitochondrial parameters were acquired and analyzed using unsupervised (hierarchical clustering) and supervised (decision tree) machine learning methods. Both biochemical and computational analyses resulted in an evident distinction between the neurotoxic effects of 6-hydroxydopamine and rotenone, specifically for the highest concentrations of both compounds.

## 1. Introduction

With the aging of the world’s population, the number of neurodegenerative disease cases has increased exponentially. It is predicted that this number will rise to around 135 million in 2050, with a projected economic burden of up to 20 billion US dollars and rising [1]. The most prominent neurodegenerative diseases, Alzheimer’s disease, Parkinson’s disease (PD), Huntington’s disease and amyotrophic lateral sclerosis, share neuropathological hallmarks such as oxidative stress, abnormal protein glycation, abnormal protein deposition (proteotoxic stress), dysfunction in ubiquitin-proteosome and autophagosomal/lysosomal systems, neuroinflammation, progressive neuronal dysfunction and cell death [2,3,4]. A key central player in the pathophysiology of neurodegenerative diseases is mitochondrial dysfunction. The central nervous system is highly dependent on this organelle to function correctly for energy supply and calcium (Ca^2+^) buffering [5,6,7]. Manifestations of mitochondrial dysfunction in neurodegenerative diseases range from altered dynamics (fusion/fission and axonal transport) leading to abnormal organelle morphology and cellular distribution, to oxidative stress, Ca^2+^ dyshomeostasis, mitochondrial membrane depolarization, decreased adenosine triphosphate (ATP) production, and mitochondrial DNA (mtDNA) damage with concomitant cell death [8,9].

No useful diagnostic tools exist for the early detection of neurodegenerative diseases, with the neuropathological evaluation at autopsy being the actual golden diagnostic standard [2]. Moreover, when clinical symptoms appear, the cellular and metabolic alterations have already reached a point of no return towards neurodegeneration and neuronal cell death. Due to the massive social and economic impact of neurodegenerative diseases, significant efforts to learn and identify risk factors and mechanisms behind the neuropathology of these diseases have been put in place [4]. One of the most promising approaches is targeting mitochondria, since the previously mentioned structural and metabolic alterations in mitochondria start at the early stages of neurodegeneration, prior to neuronal cell death [9,10]. A certain threshold of mitochondrial dysfunction and decreased ATP production must occur for visible clinical signs [11,12,13].

Differentiated SH-SY5Y cells are useful in vitro models to investigate cellular and mitochondrial morphological and bioenergetic alterations in mitochondrial dysfunction-related pathologies such as neurodegenerative diseases [14]. Moreover, 6-hydroxydopamine (6-OHDA) and rotenone have been used to develop PD models [15]. In this work, differentiated SH-SY5Y cells were treated for different time points and with distinct concentrations of 6-OHDA and rotenone, which were previously shown to affect mitochondrial movement in these cells [16]. Experimental biological analyses of cell and mitochondrial function parameters were then assessed. The combination of classical statistical methods, and unsupervised and supervised machine learning (ML) algorithms were used to assess the crosstalk between the different biological assays used, attempting to understand the differences between the mechanism of action and the degree of toxicity of 6-OHDA and rotenone. Our main objective was to rely on a ML framework to determine which experimental biological features are the most relevant in neurodegeneration induction, prior to cell death, caused by 6-OHDA and rotenone in vitro. Moreover, we wanted to unravel the different mechanisms of action of these compounds on SH-SY5Y toxicity that may help prevent neuronal cell death using different biological approaches.

## 2. Results

### 2.1. 6-OHDA and Rotenone Toxicities Were Dose and Time-Dependent

Our first objective was to choose an interval of 6-OHDA and rotenone sub-toxic concentrations, i.e., that were not cytotoxic when incubated for 24 h (Figure 1a,b, respectively) but which showed cytotoxicity at 96 h of incubation (Figure 1c,d, respectively). For this purpose, we assessed cellular metabolic viability using the resazurin assay, cell mass by the sulforhodamine B (SRB) method, ATP levels using CellTiter-Glo Luminescent Cell Viability Assay and the percentage of live cells using propidium iodide (PI) nuclei labeling, as stated in the material and methods section. The toxicity behavior of both compounds was time- and dose-dependent. When cells were treated for 96 h with increasing concentrations of 6-OHDA (0.4, 0.8, 1.6, 3.1, 6.25, 12.5, 25, 50, 100 or 200 µM), all of the referred endpoints displayed the same dose-dependent effect, showing toxicity from 6.25 µM. Cell mass only decreased for concentrations higher than 25 µM (Figure 1c). However, 6-OHDA incubated for 24 h (Figure 1a), only induced toxicity at a higher concentration (100 µM).

Regarding rotenone, and especially when cells were incubated with increasing concentrations (0.004, 0.008, 0.016, 0.03, 0.06, 0.125, 0.25, 0.5, 1 or 2 µM) for 96 h, the resulting dose–response curves were different (Figure 1d). Effects on metabolic activity and cell mass presented similar behaviors, with toxicity observed from 0.06 to 0.5 µM. Surprisingly, with higher concentrations (2 µM), the same parameters reached values similar to control. ATP levels exhibited a similar trend to the assays above, but ATP values did not increase for concentrations higher than 0.5 µM. Conversely, the percentage of live cells decreased from 0.03 µM to 2 µM of rotenone. On the other hand, when cells were incubated for 24 h with the same range of rotenone concentrations, all the parameters studied started to decrease only with 1 µM and 2 µM of the compound (Figure 1b). Based on this set of results, the concentrations chosen to investigate cellular and mitochondrial alterations prior to later cell death were 6.25, 12.5, 25 and 50 µM for 6-OHDA and 0.03, 0.06, 0.125 and 0.25 µM for rotenone (showed by the squared boxes in Figure 1) since they presented cell toxicity when incubated with these concentration intervals for 96 h but not for 24 h. We can thus consider all the measurements taken before or at 24 h as being early predictors of later neurotoxic manifestations.

For finishing the cytotoxicity profile evaluation, caspase 3/7 activity was measured with Caspase-Glo 3/7 Assay. For this, cells were treated for 24 h with the selected concentrations of both compounds or with 1 µM doxorubicin (Dox), as a positive control. The highest non-lethal concentrations of both compounds increased caspase 3/7 activity. The increase in caspase 3/7 activity was statistically significant at the highest 6-OHDA and rotenone concentrations, 50 µM and 0.25 µM, respectively (Figure 2).

### 2.2. 6-OHDA and Rotenone Decreased Mitochondrial Network Area, While Differently Affecting Mitotracker Red CMXRos Fluorescence Intensity and Mitochondrial OXPHOS and Fusion/Fission Gene Transcripts

We next investigated whether the mitochondrial network was affected by non-lethal concentrations of 6-OHDA or rotenone. Firstly, we aimed to assess alterations in the polarized mitochondrial network by high-throughput microscopy, upon labeling mitochondria with the mitochondrial membrane potential-dependent dye Mitotracker red CMXRos (Figure 3a,b). In this experiment, two time-points of incubation of 6-OHDA or rotenone were tested, 3 h (Figure 3a) and 24 h (Figure 3b). Cells treated with the higher concentrations of 6-OHDA (25 µM and 50 µM) and, interestingly, with the lower rotenone concentration (0.03 µM) for 3 h, displayed an evident decrease in Mitotracker red labeled bodies per cell, compared with untreated cells (Figure 3a). On the other hand, when cells were incubated with the two compounds for 24 h, only 6-OHDA (for all concentrations) induced a significant decrease in this parameter. At this time point, rotenone-treated cells did not present any alterations regarding this parameter compared with control cells (Figure 3b).

Since there were alterations in the polarized mitochondrial network, we next studied mitochondria membrane potential (ΔΨm), since changes in this parameter can indicate mitochondrial dysfunction. In order to indirectly address alterations in ΔΨm, we studied changes in mitochondrial network polarization by high-throughput microscopy, recurring to the previously described labeling approach. Cells were incubated with the selected concentrations of 6-OHDA or rotenone for 3 h (Figure 3c) or 24 h (Figure 3d), or with 10 µM carbonyl cyanide-p-trifluoromethoxyphenylhydrazone (FCCP), as a positive control. Cells treated with 6-OHDA showed a concentration-dependent decrease in Mitotracker Red CMXRos fluorescence intensity, especially when treated with 50 µM for 3 h (Figure 3c). When incubated for 24 h, this concentration led to a 15% decrease, on average, in dye intensity per cell, compared to control cells (Figure 3d). Rotenone-treated cells displayed an inverse effect. Cells incubated with this compound for 3 h showed increased Mitotracker Red CMXRos fluorescence intensity. The treatments with 0.125 µM or 0.25 µM showed average increases of 15% and 24% in Mitotracker Red CMXRos fluorescence intensity, respectively (Figure 3c). Regarding cells treated for 24 h, the treatment with 0.25 µM led to an evident increase in dye intensity (Figure 3d).

Confocal microscopy was used to assess more specific details regarding mitochondrial network morphology and content when cells were treated with 6-OHDA (50 µM) and rotenone (0.25 µM) for 24 h (Figure 4j–l), while data processing was performed with two ImageJ macros, namely Mito-Morphology Macro [17] and the Mitochondrial Network Analysis (MiNA) toolset [18]. With the Mito-Morphology Macro, our data show that both 6-OHDA and rotenone led to a significant decrease in the number of mitochondrial particles (Figure 4a) and maximal mitochondrial area (Figure 4b). Furthermore, rotenone caused a reduction in mitochondrial content (percentage of mitochondrial area divided by total cell area, Figure 4d). The remaining parameters (average mitochondrial area (Figure 4 c), perimeter (Figure 4e) and circularity (Figure 4f)) remained unaltered. Regarding the MiNA toolset, only 6-OHDA affected the mean number of network branches (Figure 4i), triggering an increase in this parameter. The remaining parameters (mean branch length (Figure 4g) and summed mean branch length (Figure 4k) remained unchanged in all conditions.

We next studied the expression of genes encoding proteins involved in mitochondrial fission (Appendix A) and fusion (Appendix A) after cells were treated for 24 h with 50 µM 6-OHDA or 0.25 µM rotenone to explain alterations in mitochondrial morphology. Although no statistical differences were found, both compounds seemed to induce alterations in the transcripts studied. 6-OHDA-treated cells showed an average decrease by around 60% in mRNA levels for all genes studied, except for mitochondrial elongation factor 2 (*MIEF2*) (Appendix A). Rotenone-treated cells displayed an average increase of 31% and 78% in the mRNA levels of *MIEF2* (Appendix A) and mitofusin 1 (*MFN1*) (Appendix A), respectively, when compared to control cells. On the other hand, this compound induced an average decrease of 21% in mitochondrial fission 1 protein (*FIS1*) transcripts compared to control cells (Appendix A).

### 2.3. Non-Toxic Concentrations of 6-OHDA and Rotenone Distinctly Affected Cellular and Mitochondria Bioenergetics

In order to study mitochondrial bioenergetics, the Seahorse/Agilent Mito Stress Test was used with a Seahorse XFe96 Extracellular Flux Analyzer. Here, prior to the beginning of the standard assay, cells were acutely treated with 6-OHDA (50 µM) and rotenone (0.25 µM). Unsurprisingly, basal respiration OCR was similar between the three groups (Figure 5a). However, the treatment with the two compounds had contrary effects on OCR (Figure 5a). The OCR in 6-OHDA acutely treated cells suffered, on average, a non-significant 26% increase, whereas rotenone-treated cells presented a significant decrease in OCR (78% on average) when compared to control cells (Figure 5b). A similar trend was observed in ATP production-linked OCR (Figure 5c), maximal respiration-linked OCR (Figure 5d), and spare respiratory capacity (Figure 5e). In these parameters, 6-OHDA induced, respectively, and on average, a 47%, 49%, and 35% increase in OCR, while rotenone significantly decreased the same parameters. Concerning proton leak-linked OCR, rotenone-treated cells displayed, on average, a 43% increase in this parameter, while the increase in 6-OHDA-treated cells was more evident (Figure 5f). Finally, regarding non-mitochondrial respiration-linked OCR, 6-OHDA decreased this parameter by 37% on average (Figure 5g).

This assay also records variations in ECAR (Figure 5h). As expected, the acute treatment with rotenone caused a significant increase in ECAR compared to control cells, while 6-OHDA did not affect ECAR (Figure 5i).

In addition, qPCR analyses of transcripts of electron transport chain (ETC) complexes and ATP synthase subunits encoded by nuclear DNA (Appendix A) and by mtDNA (Appendix A) were assessed after cells were treated for 24 h with the highest concentration of 6-OHDA (50 µM) or rotenone (0.25 µM) to identify alterations that could explain differences in the mitochondrial functional parameters. Although no statistical differences were found in most of the transcripts studied, both compounds seemed to induce some transcriptional alterations. 6-OHDA induced a non-significant average decrease of 37%, 30%, and 74% in nuclear-encoded ETC complex subunits NADH:Ubiquinone oxidoreductase subunit A9 (*NDUFA9*) (Appendix A), cytochrome c oxidase subunit 4I1 (*COX4I1*) (Appendix A) and ATP synthase membrane subunit c locus 1 (*ATP5G*) (Appendix A)-related transcripts, respectively, and an average decrease of 32% and 59% in mtDNA-encoded ETC complex subunits cytochrome b (*CYB*) (*p* = 0.012) (Appendix A) and ATP synthase membrane subunit 6 (*ATP6*) (Appendix A), when compared to control cells. Moreover, a significant increase was found in complex II subunit succinate dehydrogenase complex flavoprotein subunit A (*SDHA*) mRNA levels in cells treated with 6-OHDA, compared to untreated controls (Appendix A). In rotenone-treated cells, this treatment decreased 30% in nuclear-encoded ETC complex subunit *COX4I1* transcripts (Appendix A) and an average decrease of 32% and 43% in mtDNA-encoded ETC complex subunits *CYB* (Appendix A) and *ATP6* (Appendix A) mRNA levels, respectively, when compared to control cells. Furthermore, cells treated with rotenone displayed an increase of 24%, on average, in mRNA levels of the nuclear-encoded ETC complex subunit ubiquinol-cytochrome c reductase core protein 2 (*UQCRC2*) (Appendix A), when compared to untreated cells.

### 2.4. ATP Production Rate Was Disrupted by 6-OHDA and Rotenone

Next, we wanted to understand the behavior of mitochondrial and glycolytic components of real-time cellular ATP production rate. We used the Seahorse/Agilent ATP Rate assay, since it can distinguish the site of ATP production, either from glycolysis (glycoATP) or from mitochondria (mitoATP), as well as the total ATP production rate. Control cells had a substantially higher ATP production rate than treated cells after the acute treatments (Figure 6). Under control conditions, ATP production was almost equally divided from glycolytic (51%) and mitochondrial (49%) origin. The treatments decreased total ATP production differently. On the one hand, 6-OHDA (50 µM) significantly decreased both glycolytic and mitochondrial ATP production rates. However, the percentage of the respective total ATP produced from glycolysis (55%) and mitochondrial origin (45%) remained similar to control cells. On the other hand, rotenone (0.25 µM) induced a decrease in total ATP production rate that appeared to be due only to inhibition of mitochondrial-produced ATP (13% of total ATP production). With this compound, glycolytic ATP production rate appeared unaltered compared to control cells and represented 87% of total ATP produced by cells in this condition.

### 2.5. Non-Toxic Concentrations of 6-OHDA and Rotenone Triggered Intracellular Oxidative Stress

We first studied intracellular oxidative stress using the dye H2DCFDA and assessed the levels of its fluorogenic derivate 2′,7′-dichlorofluorescein (DCF). SH-SY5Y cells were incubated for 3 h (Appendix A) and 24 h (Appendix A) with the chosen concentrations of 6-OHDA and rotenone. Hydrogen peroxide (H_2_O_2_), 500 µM, was used as a positive control. No significant differences were observed in these assays with either of the test compounds. Nevertheless, a 26% and 30% increase, on average, in DCF fluorescence was measured when cells were incubated, respectively, with 0.125 µM and 0.25 µM rotenone for 3 h compared to control counterparts (Appendix A). The same treatments for 24 h induced an average increase in DCF fluorescence of 19% and 39% (Appendix A).

For further clarification of the data obtained, DCF oxidation rate was measured when cells were acutely incubated with 6-OHDA and rotenone. Remarkably, in this experiment, 6-OHDA treated cells exhibited significantly higher dye oxidation rate than their control counterparts (Figure 7a). Regardless of the concentration, rotenone-treated cells displayed an average increase of 30% in DCF oxidation rate (Figure 7a).

Next, we wanted to uncover if 6-OHDA and rotenone increased mitochondrial superoxide anion, contributing to the cellular oxidative stress. For this purpose, we used the Mitosox Red Mitochondrial Superoxide Indicator. All 6-OHDA concentrations induced a 20% increase, on average, of Mitosox oxidation rate. Rotenone concentrations 0.125 µM and 0.25 µM induced an evident increase in Mitosox oxidation rate (Figure 7c). Furthermore, at the end of the oxidation rate assay, all rotenone concentrations and 12.5 µM and 50 µM of 6-OHDA produced a significant increase in Mitosox oxidation compared to control cells (Figure 7b).

qPCR analysis of mRNA levels for antioxidant enzymes was also performed (Appendix A) with the highest concentrations of each compound. Although no statistical differences were found, both compounds seemed to induce alterations in the transcripts studied. 6-OHDA induced an overall decrease in transcripts of selected antioxidant enzymes (Appendix A) except for superoxide dismutase 1 (*SOD1*) (Appendix A). More precisely, the mRNA levels of catalase (*CAT*) were, on average, 85% decreased compared to control cells (Appendix A). On the other hand, rotenone induced an average increase of 20% in the mRNA levels of glutathione peroxidase 1 (*GPX1*) and glutathione peroxidase 4 (*GPX4*) (Appendix A respectively) and an average decrease of 30% in superoxide dismutase 2 (*SOD2*) mRNA (Appendix A).

### 2.6. 6-OHDA and Rotenone Altered the mRNA Levels of Mitochondrial Biogenesis-Associated Genes While Decreasing mtDNA Copy Number

Since both 6-OHDA and rotenone induced alterations in mitochondria morphology and bioenergetics, we next aimed to assess whether these treatments affected mtDNA copy number and mRNA levels of mitochondrial biogenesis-associated genes. Although we did not find any statistical difference in mtDNA copy number induced by any compound, a 36% and 43% average decrease in mtDNA copy number was measured when cells were treated with 50 µM 6-OHDA or 0.25 µM rotenone for 24 h, respectively (Figure 8a).

Next, we assessed the mRNA levels of the transcription factors nuclear respiratory factor 1 (*NRF1*) (Figure 8b), mitochondrial transcription factor A (*TFAM*) (Figure 8c) and GA binding protein transcription factor subunit beta 1 (GABPB1) (Figure 8d), and the enzymes sirtuin 1 (*SIRT1*) (Figure 8e) and adenosine monophosphate-activated protein kinase (*AMPK*) (Figure 8f), shown to play pivotal roles in mtDNA replication and transcription, and concomitantly in mitochondrial biogenesis [19]. Regarding the transcription factors, both 6-OHDA (50 µM) and rotenone (0.25 µM), incubated for 24 h, induced a decrease in *NRF1* (Figure 8b) and *TFAM* (Figure 8c) mRNA levels. More precisely, rotenone caused a significant decrease in *NRF1* transcripts, while 6-OHDA also prompted, on average, a 47% decrease in mRNA levels of *NRF1* (*p* = 0.06) (Figure 8b) and a 65% decrease in TFAM mRNA levels (*p* = 0.08) (Figure 8c). Furthermore, although 6-OHDA did not alter *GABPB1* mRNA levels, 24 h incubation with rotenone significantly increased the transcript levels of this transcription factor (Figure 8d). On the other hand, concerning the mRNA levels of genes involved in mitochondrial biogenesis, only rotenone treatment had a marked effect, increasing the mRNA levels of both *AMPK* (Figure 8e) and *SIRT1* (Figure 8f). 6-OHDA did not induce any effects in the mRNA levels of the same genes.

### 2.7. 6-OHDA and Rotenone Prompted Lysosomal Protease Activation and an Increase in Lysosomal Area and Number

Since we showed a decrease in mitochondrial polarization and an increase in cellular and mitochondrial-derived oxidative stress, we next aimed to study mitophagy-related events, i.e., a quality control mechanism aimed at removing dysfunctional mitochondria through lysosomal degradation [20], such as lysosomal protease activity and lysosomal-related parameters such as lysosomal number, area per cell and average area.

The measurement of lysosomal protease activity was performed using the fluorogenic substrate DQ Green BSA. The hydrolysis rate of this substrate was measured after acute treatment with increasing concentrations of 6-OHDA and rotenone. All 6-OHDA concentrations except, unexpectedly, 25 µM showed a significant increase in lysosomal protease activity. Only cells treated with the highest concentrations of rotenone, 0.125 µM and 0.25 µM, displayed a significant increase in DQ Green BSA hydrolysis (Figure 9a).

Additionally, we also studied (by confocal microscopy, Figure 9e–g) lysosome-related parameters such as lysosome number (Figure 9b) and area (Figure 9c) per cell, as well as lysosomal average area (Figure 9d). Rotenone treatment (0.25 µM) induced an average increase of 42% in the lysosomal area per nuclei (Figure 9d) and a significant increase in the number of lysosomes, when compared to control cells (Figure 9b). 6-OHDA treatment did not cause any alterations in the studied lysosomal parameters.

### 2.8. Eight Out of 11 Features with Information Gain Higher than 0.5 Provided a Good Separation between Experimental Groups

The differences between the mechanisms and degree of toxicity of 6-OHDA and rotenone were investigated using computational applied statistical methods. As stated in the materials and methods section, the complete dataset for this analysis comprised data obtained in control cells and in cells treated either with the four concentrations of 6-OHDA (6.25, 12.5, 25 or 50 µM) or with the four concentrations of rotenone (0.03, 0.06, 0.125 or 0.25 µM). This includes 36 samples, each described by 11 numeric features and one target (a specific cell treatment). These 11 numeric features encompass experimental assays done before or at 24 h of compound incubation, being considered early predictors of later neurotoxic manifestations.

To understand which subset of features measured in this work contributed to a better discrimination between the experimental groups, we determined the mutual information (information gain) between each individual feature and the experimental class. A subset of 8 out of 11 features (Mitotracker red intensity 3 h and 24 h, Mitotracker red area 3 h and 24 h, Caspase Glo 24 h, Mitosox oxidation 3 h and oxidation rate, and DCFDA fluorescence 3 h) had information gains higher than 0.5 (Figure 10a). Hierarchical clustering using this subset of features was then applied to understand if we could distinctly cluster control cells and cells treated with either 6-OHDA or rotenone, aiming at better understanding the different mechanisms and degree of toxicity of both compounds (Figure 10b).

Hierarchical clustering perfectly separated control cells from cells treated with different concentrations of both treatments (6-OHDA and rotenone). After the clustering step, leaves were labelled with the corresponding concentrations (Figure 10b). When the dendrogram was divided into different clusters, we could isolate control cells (red box, C1). A different cluster (Black box, C2) grouped cells treated with higher 6-OHDA concentrations (25 µM and 50 µM). Cluster C3 (green box) contained cells treated with different rotenone concentrations. The remaining clusters grouped cells treated either with 6-OHDA or rotenone. Interestingly, these clusters contained cells treated with lower concentrations of both compounds, shown to induce small and similar effects.

### 2.9. A Decision Tree Model Presented High Predictive Power for the Highest Concentration of Both Compounds and for the Lowest Concentration of 6-OHDA

To corroborate that (i) the unsupervised analysis and the biological findings showing that 6-OHDA and rotenone have different mechanisms of action and degrees of toxicity, and (ii) that this toxicity was time- and dose-dependent, a supervised decision tree model was trained to predict the real class from the data. The test results obtained using the decision tree are shown in a confusion matrix in Table 1. This model was only able to classify 50% of the control samples correctly. The remaining 50% were predicted to be either cells treated with 6.25 µM 6-OHDA (25%) or cells treated with 0.25 µM rotenone (25%). If the first value could be anticipated since lower 6-OHDA concentrations and control cells presented several similar biological effects, the second value (control cells predicted as cells incubated with 0.25 µM rotenone) was unexpected since this concentration was the highest rotenone concentration and displayed distinct biological effects in comparison to control cells. This algorithm performed well by correctly predicting cells incubated with the highest (50 µM) and the lowest (6.25 µM) 6-OHDA concentrations. In both cases, the percentage of correctly classified samples was 75%. In the case of cells treated with 50 µM 6-OHDA, the remaining 25% (which represents just one sample) of misclassified labeling was predicted to be cells treated with 25 µM 6-OHDA, which could be expected since this concentration was the second highest one, and biologically displayed very similar effects when compared to cells treated with 50 µM 6-OHDA. On the other hand, and unexpectedly, the 25% of misclassified 6.25 µM 6-OHDA treated samples (it is worth noting that this percentage represents just one sample) were classified as cells treated with 0.125 µM rotenone. One explanation could be that lower 6-OHDA concentrations could display similar effects to this rotenone concentration.

Cells treated with 0.25 µM rotenone were entirely correctly classified, while the classification of cells treated with 0.125 µM was only 50% correct. The remaining samples treated with this rotenone concentration were predicted to be either 0.25 µM (25%) or 0.06 µM (25%) rotenone, which makes sense since those were the two closest rotenone concentrations.

The remaining class predictions had low values (25%). In the case of 6-OHDA-treated cells, the misclassified samples appeared in the closest 6-OHDA concentrations. Interestingly, cells treated with the lowest rotenone concentrations (0.03 and 0.06 µM) both had 25% of samples predicted to belong to the 6.25 µM 6-OHDA. This could be because the lowest concentration of both compounds could cause similar effects on SH-SY5Y cells.

## 3. Discussion

The use of computational applied statistical methodologies in biology, i.e., combining classical statistics with ML algorithms, is a continuously evolving field. In this interdisciplinary work, we applied ML methodologies in original biological data to predict which experimental features could inform us about the mechanism and degree of toxicity of 6-OHDA and rotenone related with neurodegeneration before cell death. Concerning the ML approaches, we used unsupervised and supervised methods. Finally, we performed an interdisciplinary analysis of both biological and computational results retrieving the most important conclusions from the combination of both fields.

Different studies revealed similar results regarding the effects of 6-OHDA and rotenone on SH-SY5Y cells. Firstly, both compounds were previously shown to induce a mitochondrial fragmentation phenotype [21,22,23] and prompt a loss of ΔΨm, release of cytochrome c from mitochondria, reduction of mtDNA content, and decreased protein levels of PGC-1α, NRF1, and TFAM [22,24]. Additionally, increased oxidative stress and caspase 3 activity were also previously found in SH-SY5Y treated with these compounds [21,25,26,27]. Using different cell models, both compounds were previously shown to induce mitochondrial bioenergetic deficits. Using rat forebrain mitochondria, 6-OHDA induced a reduction in respiratory control ratio and rate of oxygen consumption at state 3 (state 4 remained unaltered) [28], while primary rat cortical neurons treated with rotenone presented an evident decrease in OCR-linked ATP production, maximal respiration-linked OCR, and spare respiratory capacity, while non-mitochondrial respiration-linked OCR remained unaltered [29].

Additionally, both compounds have been reported to induce autophagy. A study by Chu and collaborators, using both primary cortical neurons and undifferentiated SH-SY5Y treated with sublethal concentrations of rotenone and 6-OHDA, showed an evident increase in LC3 levels and LC3 puncta co-localized with mitochondria. A decrease in p62 levels was also present, suggesting an increased autophagic flux [30].

Nonetheless, 6-OHDA and rotenone are mitochondrial toxicants known to have different mechanisms of action. Firstly, while rotenone is a highly lipophilic and hydrophobic molecule that can easily cross cell membranes without the need for specific membrane transport mechanisms [31], 6-OHDA cellular uptake needs specific transporters such as the dopamine or norepinephrine transporters [32,33]. Regarding mitochondrial-induced toxic effects, these compounds also display different mechanisms of action. Rotenone is a golden-standard high-affinity, time-dependent, irreversible mitochondrial complex I inhibitor [34] while 6-OHDA inhibits both mitochondrial complexes I and IV [35]. Complex I inhibition by both compounds leads to superoxide production [15,36]. Moreover, 6-OHDA is an easily oxidizable molecule, forming semiquinone radicals, participating in free radical producing reactions and leading to H_2_O_2_, O_2_^• −^ and ^•^OH production [37].

These differences could help to explain why, in our work, despite inducing similar effects in caspase activity, mitochondrial-encoded ETC subunits mRNA levels, mitochondrial morphology, Mitosox Red oxidation, mtDNA copy number, and lysosomal protease activity; in the remaining parameters, 6-OHDA and rotenone promoted distinct effects. 6-OHDA induced a decrease in mitochondrial area and ΔΨm, mRNA levels of nuclear-encoded ETC subunits, fusion/fission proteins, and antioxidant enzymes, and ATP production rate. Moreover, 6-OHDA also prompted an acute increase in mitochondrial bioenergetic parameters. Rotenone promoted a decrease in mitochondrial bioenergetic parameters and ATP production rate while inducing an increase in ΔΨm and mitochondrial biogenesis-associated genes.

To understand the differences and similarities of non-toxic concentrations of 6-OHDA and rotenone on differentiated SH-SY5Y cells and to complement our biological data analysis, ML methodologies, such as hierarchical clustering and supervised classification algorithms, were used to clarify two main questions. Firstly, was it possible to distinguish between control and treated cells? And if so, is it also possible to distinguish between control cells and cells treated with different 6-OHDA or rotenone concentrations? Hierarchical clustering is an unsupervised approach that estimates the dissimilarities (based on distances) between any pair of observations. Being an unsupervised method, hierarchical clustering aimed to discover hidden patterns in the data without using the respective class labels for each observation. This methodology allowed us to have a broader idea of how the different observations were organized. Information gain revealed that the features MitoTracker area and intensity (3 and 24 h), mitochondrial (Mitosox oxidation 3 h and oxidation rate) and cellular (DCFDA fluorescence 3 h)-derived oxidative stress, and Caspase activity, seem to have an enhanced ability to signal neurodegeneration induction, prior to cell death, caused by 6-OHDA and rotenone. Using this subset of eight features with higher information gain, hierarchical clustering automatically grouped control cells in one cluster and cells treated with higher concentrations of 6-OHDA and higher rotenone concentrations in distinct clusters. This occurred because the higher the concentration of each compound, the more evidently different biologic effects on SH-SY5Y cells were demonstrated. On the other hand, milder rotenone and 6-OHDA concentrations were clustered together since they caused smaller and similar effects. Biological results were also used to train a decision tree predictive model. In this methodology, the output labels are known during model training (contrary to hierarchical clustering). The decision tree algorithm could correctly classify a high percentage (>75%) of cells treated with the highest 6-OHDA and rotenone concentrations (50 µM and 0.25 µM, respectively), but only 50% of control cells. Interestingly, this model was able to predict correctly 75% of cells treated with 6.25 µM 6-OHDA, being an improvement from the results derived from the hierarchical clustering analysis. Regarding cells treated with the lowest rotenone concentration, the hierarchical clustering results already suggest that classification methods may be unable to classify samples in their respective class correctly. Misclassified samples were predicted to be in classes corresponding to the remaining rotenone concentrations or the class of cells treated with 6.25 µM 6-OHDA. Similar to what happened in the hierarchical clustering analysis, the predictive model was unable to distinguish between cells treated with a milder concentration of both compounds (except on cells treated with 6.25 µM 6-OHDA), since the biological effects caused by these conditions only induced smaller and similar effects.

## 4. Materials and Methods

### 4.1. Cell Culture

SH-SY5Y human cell line (ECACC, cat. 94030304, Merck KGaA, Darmstadt, Germany) was cultured in a humidified atmosphere with 5% CO_2_, 95% air at 37 °C. The media used for undifferentiated cells was Dulbecco’s Modified Eagle’s Medium (DMEM D5030, Sigma-Aldrich, Merck KGaA, Darmstadt, Germany) supplemented with 4.5 g/L D-(+)-glucose (cat. G7021, Sigma-Aldrich, Merck KGaA, Darmstadt, Germany), 0.8 g/L L-glutamine (cat. G3126, Sigma-Aldrich, Merck KGaA, Darmstadt, Germany), 1.2 g/L HEPES (cat. H4024, Sigma-Aldrich, Merck KGaA, Darmstadt, Germany), 3.7 g/L sodium bicarbonate (cat. S6014, Sigma-Aldrich, Merck KGaA, Darmstadt, Germany), 0.1 g/L sodium pyruvate (cat. P2256, Sigma-Aldrich, Merck KGaA, Darmstadt, Germany), 10% fetal bovine serum (FBS, cat. 41F6445K, Gibco, Thermo Fisher Scientific, Hampton, NH, USA) and 1% penicillin/streptomycin (cat 1772652 Thermo Fisher Scientific, Hampton, NH, USA). Cell media was changed every 2 to 3 days and cells were subcultured, until the 23rd passage, when reaching 90–100% of confluence.

We previously described the protocol used for SH-SY5Y cell differentiation in ref. [14]. Briefly, undifferentiated cells were plated at a density of 30,000 cells/cm^2^ in a low D-(+)-glucose (0.9 g/L) media supplemented with 1% FBS and 10 µM retinoic acid (cat. A6947 Panreac AppliChem ITW Reagents, Darmstadt, Germany) for 3 days. Exceptionally, cells were plated at a density of 120,000 cells/cm^2^ in the Seahorse experiments (Seahorse XF Cell Mito Stress Test and Real-Time ATP Rate Assay) due to equipment sensitivity.

### 4.2. Cell Treatments

Two mitochondrial poisons were used, 6-OHDA (cat. H4381 Sigma-Aldrich, Merck KGaA, Darmstadt, Germany) and rotenone (Cat. MKBS1062V, Sigma-Aldrich, Merck KGaA, Darmstadt, Germany). Rotenone was dissolved in dimethyl sulfoxide (DMSO, Thermo Fisher Scientific, Hampton, NH, USA) while 6-OHDA is water-soluble. The final concentration of DMSO used was never higher than 0.1% (*v*/*v*). Cell treatment strategies are specified in the respective results subsection and in the respective figure legend.

### 4.3. Cell Proliferation Measurements by Sulforhodamine B Method

To evaluate the cytotoxic effects of 6-OHDA and rotenone on cell mass, the sulforhodamine B (SRB) assay [38] was used to indirectly evaluate the total cell protein [39]. After cell treatments, the medium was removed, and cells were fixed in 1% acetic acid in ice-cold methanol overnight at −20 °C. Then, cells were incubated with 0.05% (*w*/*v*) SRB (Cat. S9012, Sigma-Aldrich, Merck KGaA, Darmstadt, Germany) for 1 h at 37 °C. Subsequently, SRB was removed, and wells were washed with 1% acetic acid to remove the unbound stain. Dye bound to cell proteins was extracted with 10 mM Tris-base solution, pH 10, and absorbance was read at 510 nm and 620 nm (reference wavelength) in a Cytation 3 reader (BioTek Instruments Inc., Winooski, VT, USA). Then, we subtracted the 620 nm absorbance value from the 510 nm absorbance value for each condition. Data were normalized to the average control (100%) condition.

### 4.4. Cellular Metabolic Viability Assessment Using the Resazurin Assay

The effect of 6-OHDA and rotenone toxicity was also evaluated by measuring the cellular metabolic activity using the resazurin reduction assay. The resazurin assay is based on the reduction of resazurin (Cat. R7017-5G, Sigma-Aldrich, Merck KGaA, Darmstadt, Germany) to resorufin by dehydrogenases present in viable cells [39]. After cell treatments, the medium was replaced by a fresh medium containing resazurin (10 μg/mL) and kept in a humidified atmosphere, 5% CO_2_, at 37 °C for 3 h. The fluorescent signal was monitored using a 540 nm excitation wavelength and 590 nm emission wavelength in a Cytation 3 microplate reader. A blank condition (wells without cells) was used as a fluorescent reference signal and subtracted from the remaining conditions. Data were normalized to the average control (100%) condition.

### 4.5. ATP Levels Determination

Intracellular ATP levels were assessed using CellTiter-Glo Luminescent Cell Viability Assay (cat. G7570, Promega, Madison, WI, USA). Cell differentiation and treatments were undertaken in 96-well white opaque-bottom plates (cat. 136101, Thermo Fisher Scientific, Hampton, NH, USA) and, at the end of the cell treatments, the medium was removed from the wells, replaced by 50 µL of fresh medium plus 50 µL of the Cell Titer-Glo reagent, and mixed for 2 min on an orbital shaker to promote cell lysis. The plate was incubated for 10 min and the luminescent signal was recorded using a Cytation 3 microplate reader (BioTek Instruments Inc., Winooski, VT, USA). Data were normalized to the average control (100%) condition.

### 4.6. Live Cell Calculation

Cell membrane integrity and cell death induction were evaluated using propidium iodide (PI) and Hoechst 33,342 staining. After the differentiation protocol and 6-OHDA and rotenone incubations, cell nuclei were stained with 4 µM PI (cat. P21493, Thermo Fisher Scientific, Hampton, NH, USA) and 1 µg/mL of Hoechst 33342 (cat. B2261, Sigma-Aldrich, Merck KGaA, Darmstadt, Germany). For being a cell-impermeant stain, only cells without an intact membrane were labeled with PI, while Hoechst 33342 labeled all cell nuclei. Both PI and Hoechst 33,342 were added directly to the media without aspirating it in order to avoid removing dead, PI-positive, cells. Cells were incubated with both stains for 5 min and visualized using an INCell Analyzer 2200 (GE Healthcare, Chicago, IL, USA) cell imaging system with 10× magnification (Nikon 10×/0.45, Plan Apo, CFI/60, Tokyo, Japan). The image analysis was performed using IN Cell Analyzer 1000 analysis Software-Developer Toolbox (GE Healthcare, Chicago, IL, USA). The PI negative cells were divided by the total number of cells to calculate the live cell population. Data were normalized to the average control (100%) condition.

### 4.7. Caspase 3/7 Activity

Caspase 3/7 activity was measured using the Caspase-Glo 3/7 Assay (Promega, Madison, WI, USA). Cell treatments were performed in 96-well white opaque-bottom plates. Some cells were treated with 1 µM Doxorubicin (Cat. BP2516, Thermo Fisher Scientific, Hampton, NH, USA) for 24 h as a positive control. At the time of the assay, the cell medium was removed from the wells and replaced by 50 µL of fresh medium. Then, 50 µL of the Caspase-Glo 3/7 reagent was added and mixed for 2 min on an orbital shaker to promote cell lysis. The plate was incubated for 30 min and the luminescent signal was recorded using a Cytation 3 microplate reader. A blank condition (wells without cells) was used as a luminescent reference signal and subtracted from the remaining conditions.

### 4.8. Mitochondrial Polarization and Network Distribution Analysis by Live Cell High Throughput Microscopy

Live cell high-throughput microscopy was used to detect alterations in mitochondrial polarization and network distribution in differentiated SH-SY5Y cells exposed to increasing (non-toxic) concentrations of 6-OHDA and rotenone. Cells were differentiated on Corning 96-well black polystyrene microplates (Cat. 3603, Corning GmbH, Kaiserslautern, Germany) and after the treatment with either compound, mitochondria were stained for 30 min with 50 nM of the mitochondrial fluorescent dye Mitotracker Red CMXRos (Cat. M7512, Thermo Fisher Scientific, Hampton, NH, USA) in a humidified atmosphere, 5% CO_2_, at 37 °C. For positive control, 10 µM carbonyl cyanide-4-(trifluoromethoxy) phenylhydrazone (FCCP, Cat.024M4003V, Sigma-Aldrich, Merck KGaA, Darmstadt, Germany) was used and incubated at the same time as the dye incubation. Cells were also stained with 1 µg/mL Hoechst 33,342 for normalization purposes. An INCell Analyzer 2200 (GE Healthcare, Chicago, IL, USA) cell imaging system was used for visualization and image acquisition using a 20× objective (INCA ASAC 20×/0.45, ELWD Plan Fluor). The image analysis was performed using the IN Cell Analyzer 1000 analysis software-Developer Toolbox. A protocol was created to quantify a specific target’s fluorescence in IN Cell Developer Toolbox for image analysis. The image stacks were uploaded to the program to identify our target set and establish the respective parameters: area, intensity, and number. The representative images were visualized using ImageJ 1.52a (Wayne Rasband, National Institute of Health, Bethesda, MD, USA).

### 4.9. Mitochondrial Network Evaluation and Lysosomal Analysis by Confocal Microscopy

Confocal microscopy was used to accomplish a more detailed definition of the mitochondrial network, to be able to apply the ImageJ macros Mitochondrial Network Analysis (MiNA) toolset [18] and Mito-Morphology Macro [17], and to study lysosomal parameters. For mitochondrial network analysis, cells were differentiated in µ-slide 8 well chambers (Cat. 80826, Ibidi GmbH, Gräfelfing, Germany) and subsequently treated with either 50 µM 6-OHDA or 0.25 µM rotenone for 24 h. After these treatments, cells were incubated for 30 min with 50 nM of the mitochondrial fluorescent dye Mitotracker Red CMXRos in a humidified atmosphere, 5% CO_2_, at 37 °C. Subsequently, cells were washed 3 times with phosphate buffer saline (PBS) 1× and fixed with 4% paraformaldehyde in PBS 1× overnight at 4 °C. Afterwards, cells were washed 3 times with PBS 1x and incubated with 1 μg/mL Hoechst 33342 in PBS 1× to label cell nuclei. Regarding the study of lysosomal parameters, after cell differentiation and treatment, cells were incubated for 30 min with 75 nM LysoTracker Green DND-26 and 1 μg/μL Hoechst 33,342 in a humidified atmosphere, 5% CO_2_, at 37 °C. Finally, cells were imaged under a Carl Zeiss Laser scanning confocal microscope (LSM 710, Carl Zeiss, Jena, Germany) using an oil Plan-Apochromat 63× DIC M27 (Carl Zeiss, Jena, Germany) objective with a numerical aperture of 1.4. A BP 545/25, FT 570, BP 605/70 filter set was used to measure the red fluorescence from Mitotracker Red CMXRos, a BP 470/40, FT 495, BP 525/50 for the measurement of the green fluorescence from LysoTracker Green DND-26 and a G 365, FT 395, BP 445/50 filter set was used to assess the blue fluorescence from Hoechst 33342. Excitation was achieved with an HXP 120V light source. For fixed cells mitochondrial analysis, Z-stack series were made and consisted of 0.2–0.4 µm slice intervals and rendered into a single 2D images using the “maximum intensity projection” processing tool using Zeiss Zen 2 software (Carl Zeiss, Jena, Germany).

By using ImageJ 1.52a, it was possible to apply the MiNA toolset that provided us with data related to the mean of mitochondrial branch length, summed mitochondrial branch length and mitochondrial network branches [18]. Moreover, with the Mito-Morphology Macro, we obtained parameters such as the number of mitochondrial particles, mitochondria maximal area, average size, content, perimeter, and circularity [17].

### 4.10. Cellular Oxygen Consumption Rate (OCR) and Extracellular Acidification Rate (ECAR) Measurements

Oxygen consumption and extracellular acidification rates were measured at 37 °C using a Seahorse XFe96 Extracellular Flux Analyzer (Agilent Technologies, Santa Clara, CA, USA) following the Seahorse XF Cell Mito Stress Test (cat. 103015-100, Agilent Technologies, Santa Clara, CA, USA) and Real-Time ATP Rate (cat. 103592-100, Agilent Technologies, Santa Clara, CA, USA) Assay protocols. To be in the range of the equipment sensitivity, cells were differentiated at a density of 120,000 cells/cm^2^. Firstly, a XFe96 sensor cartridge was placed in a 96-well calibration plate containing 200 µL/well of calibration buffer and left to hydrate overnight at 37 °C. Next, the medium in the 96-well plate containing the cells was replaced with 175 µL/well of prewarmed low-buffered serum-free minimal DMEM (D5030, Sigma-Aldrich, Merck KGaA, Darmstadt, Germany) medium (pH 7.4) and incubated at 37 °C for 1 h to allow the temperature and pH of the medium to reach equilibrium before the first measurement. For the Seahorse XF Cell Mito Stress Test assay measurement of OCR and ECAR, 2 µM oligomycin (Cat. SLBS6501, Sigma-Aldrich, Merck KGaA, Darmstadt, Germany) were injected through port B, 2 µM FCCP were injected through port C and port D was loaded with 1 µM rotenone (Cat. MKBS1062V, Sigma-Aldrich, Merck KGaA, Darmstadt, Germany) plus 1 µM antimycin A (Cat. 036M4041V, Sigma-Aldrich, Merck KGaA, Darmstadt, Germany). Both the calibration plate and the sensor cartridge were loaded into the XFe96 Extracellular Flux Analyzer for calibration. The calibration plate was then replaced with the study plate. Three baseline rate measurements of SH-SY5Y cell line OCR were made using a 3 min mix, 3 min measure for 3 cycles. Rotenone (0.25 µM), 6-OHDA (50 µM) or medium (control) were then pneumatically injected through port A into the corresponding wells and mixed, with OCR measurements made using a 3 min mix, 3 min measuring time for 5 cycles. Lastly, oligomycin, FCCP, rotenone and antimycin A were sequentially pneumatically injected and mixed, with OCR measurements made using a 3 min mix, 3 min measuring time for 3 cycles. Several parameters were evaluated, such as the acute response, ATP production, proton leak, maximal respiration, spare respiratory capacity and non-mitochondrial respiration-linked OCR. Acute responses were measured as the last OCR measurement before oligomycin injection minus the last measurement before port A injection. After adding 2 µM oligomycin, it was possible to calculate ATP production-linked OCR by subtracting the last measurement before oligomycin injection and the minimum rate measurement after the oligomycin injection. Moreover, it was also possible to calculate the proton leak-linked OCR by subtracting the non-mitochondrial respiration-linked OCR from the minimum rate measurement after oligomycin injection. When cells were treated with 2 µM FCCP it was possible to evaluate the maximal respiration-linked OCR and spare respiratory capacity. Maximal respiration-linked OCR corresponds to the maximum rate measurement after FCCP injection minus the non-mitochondrial respiration-linked OCR, while the spare respiratory capacity is calculated by subtracting the basal from the maximal respiration. Lastly, non-mitochondrial respiration-linked OCR is the minimum rate measurement after rotenone and antimycin A injection.

Regarding the Real-Time ATP Rate Assay, the procedure was very similar. However, in this assay, the cells were not injected with FCCP. The cells were firstly acutely treated with rotenone (0.25 µM), 6-OHDA (50 µM) or medium (control) through port A. Subsequently, cells were treated with 2 µM oligomycin through port B followed by 1 µM rotenone plus 1 µM antimycin A through port C. The cycles of mix and measuring were the same used in the Mito Stress Test assays. Oligomycin, FCCP, rotenone, and antimycin A were prepared in DMSO and diluted to the desired final concentration in serum-free minimal DMEM.

For both assays, results were normalized by cell nuclei number in each well and analyzed using the Software Version Wave Desktop 2.6 (Agilent Technologies, Santa Clara, CA, USA).

### 4.11. Evaluation of Intracellular Oxidative Stress

For studying the effect of 6-OHDA and rotenone on intracellular oxidative stress, the cell-permeant 2′,7′-dichlorodihydrofluorescein diacetate (H2DCFDA, cat. C6827, Molecular Probes, Invitrogen, Eugene, OR, USA) dye was used. Two types of assays were carried out, endpoint and kinetic assays. For the endpoint assay, cells were differentiated in Corning 96-well black polystyrene microplates and subsequently treated with increasing concentrations of 6-OHDA, rotenone or 500 µM of hydrogen peroxide (H_2_O_2_, cat. BCBT5298 Panreac AppliChem ITW Reagents, Darmstadt, Germany) as positive control for 3 h or 24 h. At the end of these incubation periods, cells were incubated with 5 µM of H2DCFDA for 30 min at 37 °C, 5% CO_2_, and the fluorescent signal was monitored using a 485 nm excitation wavelength and 528 nm emission wavelength in a Cytation 3 reader. Regarding the kinetic assay, the differentiation process was the same. Then, cells were incubated with 5 µM H2DCFDA for 30 min at 37 °C, 5% CO_2_. After this, the medium containing the dye was removed, and cells were treated with increasing concentrations of 6-OHDA, rotenone or 1 mM H_2_O_2_ as a positive control. Lastly, the fluorescent signal was monitored using a 485 nm excitation wavelength and 528 nm emission wavelength in a Cytation 3 reader. A kinetic assay was made measuring the fluorescent signal every minute for 5 h in order to determine where dye oxidation was linear. This read always occurs in the first 50 min. Then, a linear regression analysis was calculated, and the slope was used to estimate the H2DCFDA oxidation rate. In these experiments, low D-(+)-glucose medium supplemented with 1% FBS and 10 µM RA but without sodium bicarbonate was used.

### 4.12. Determination of Mitochondrial Superoxide Anion-Dependent Oxidative Stress

In order to determine if oxidative stress had a mitochondrial contribution, the fluorogenic dye Mitosox Red Mitochondrial Superoxide indicator (cat. 1626551, Thermo Scientific, Waltham, MA, USA) was utilized. This dye is specifically targeted to mitochondria in live cells [40]. After cell differentiation in Corning 96-well black polystyrene microplates, cells were incubated with 5 µM Mitosox for 30 min at 37 °C, 5% CO_2_. Then, and without removing the medium containing the dye, cells were treated with increasing concentrations of 6-OHDA, rotenone or 2 µM antimycin A as positive control. After this, the fluorescent signal was monitored using a 510 nm excitation wavelength and 580 nm emission wavelength in a Cytation 3 reader. A kinetic assay was made by measuring the fluorescent signal every 2 min for 4.5 h in order to determine where ROS production was linear. This occurred between 120 and 200 min. Then, a linear regression analysis was calculated, and the slope was used to estimate Mitosox oxidation rate. Moreover, the value of the fluorescent signal at the final time point was also used as an endpoint assay. In these experiments, low D-(+)-glucose medium supplemented with 1% FBS and 10 µM RA but without sodium bicarbonate was used.

### 4.13. Calculation of mtDNA Copy Number

After cell differentiation and treatments, trypsinized-collected cells were centrifuged at 2000× *g* for 5 min. The pellets were washed in 5 mL of PBS 1× and the suspension was again centrifuged at 2000 g for 5 min. The resulting pellets were stored at −80 °C until total DNA extraction. Total DNA was extracted from cell pellets using the QIAamp DNA mini kit (Qiagen, Dusseldorf, Germany), following the manufacturer’s protocol. The samples were sonicated for 10 min to avoid dilution bias [41] and then DNA concentration was quantified using a Nanodrop 2000 (Thermo Scientific, Waltham, MA, USA). Real time-PCR was performed using the SsoFast Eva Green Supermix, in a CFX96 real-time PCR system (Bio-Rad, Hercules, CA, USA), with the primers described in Table 2, at 500 nM. Amplification of 25 ng of total DNA was performed with an initial cycle of 2 min at 98 °C, followed by 40 cycles of 5 s at 98 °C plus 5 s at 60 °C, and 5 s at 65 °C. At the end of each cycle, Eva Green fluorescence was recorded to enable determination of Cq. For each set of primers, amplification efficiency was assessed. To evaluate the reaction efficiency, a standard curve using serial dilutions of a representative sample was generated and then the efficiency for a subsequent gene expression analysis was recorded. The efficiency of the PCR reactions was between 95–105%. mtDNA copy number was determined in each sample by the ratio between the amount of a fragment of the mitochondrial cytochrome B (mito CyB) and the amount of the beta-2-microglobulin (B2m) nuclear gene, using the CFX96 Manager software (v. 3.0; Bio-Rad, Hercules, CA, USA).

### 4.14. Gene Expression Assessment

PureZOL Reagent (Cat.7326880, Bio-Rad, Hercules, CA, USA) was used to isolate RNA fractions from whole-cell samples. In order to do this, cells were differentiated in 6-well plates and then treated either with 50 µM 6-OHDA or 0.25 µM rotenone for 24 h. After this, the medium was removed and 1 mL of PureZOL was added to each well. The reagent was mixed by pipetting up and down and incubated at room temperature for 5 min to allow nucleoprotein complexes disruption. Total RNA was extracted with RNeasy mini kit (Qiagen, Dusseldorf, Germany), following the manufacturer’s protocol. Total RNA concentration and purity were evaluated using a Nanodrop 2000 (ThermoScientific, Waltham, MA, USA). Subsequently, RNA was converted into cDNA using the iScript cDNA synthesis kit (Cat. 170-8890, Bio-Rad, Hercules, CA, USA), following the manufacturer’s instructions. Quantitative reverse-transcriptase PCR (qRT-PCR) was performed using the SsoFast Eva Green Supermix, in a CFX96 real-time PCR system (Bio-Rad, Hercules, CA, USA), with the primers defined in Table 2, at 500 nM. Amplification of 25 ng of DNA was performed with an initial cycle of 30 s at 95 °C, followed by 40 cycles of 5 s at 95 °C plus 5 s at 60 °C. At the end of each cycle, Eva Green fluorescence was recorded to enable determination of Cq. For each set of primers, amplification efficiency was assessed. For evaluating the reaction efficiency, a standard curve using serial dilutions of a representative sample was generated and then the efficiency for subsequent gene expression analysis was recorded. The efficiency of the PCR reactions was between 95–105%. Relative normalized expression was determined by the CFX96 Manager software (v. 3.0; Bio-Rad, Hercules, CA, USA), using B2M and TATA-Box Binding Protein (TBP) as reference genes (M value of 0.901 and Stability of 0.101).

### 4.15. Measurement of Lysosomal Protease Activity

Lysosomal protease activity in cells was measured using DQ Green BSA (cat. D12050, ThermoScientific, Waltham, MA, USA). When this quenched fluorogenic substrate suffers hydrolysis by cellular proteases the quenching disappears, and a bright product is produced. After cell differentiation in Corning 96-well black polystyrene microplates, the medium was replaced with new medium containing 10 µg/mL DQ Green BSA and incubated for 1 h at 37 °C. After dye incubation, cells were treated with increasing concentration of 6-OHDA, rotenone, or 2 µM trichostatin A (TSA, cat. T8552 Sigma-Aldrich, Merck KGaA, Darmstadt, Germany) as positive control in Earle’s Balanced Salt Solution (EBSS) medium. Lysosomal protease activity was measured as fluorescence in a Cytation 3 reader at 495 nm excitation, 525 nm emission each min for 4.5 h in order to find out where DQ Green BSA hydrolysis was linear. This linearity occurred always in the last 170 min. Then, a linear regression analysis was calculated, and the slope was used to estimate the DQ Green BSA hydrolysis rate.

### 4.16. Applied Statistical Analysis

The complete dataset used in the analysis comprises 36 samples, each described by 11 numeric features and 1 target. The attributes considered were caspase 3/7 activity, Mitotracker Red CMXRos area and intensity (3 h and 24 h incubations with both compounds), Mitosox oxidation (3 h incubation with the referred compounds) and oxidation rate, DCFDA fluorescence (3 h and 24 h incubations with either compound) and oxidation rate, and DQ BSA hydrolysis. The target of each instance corresponds to one of the 9 possible classes (4 samples per class): Control, 6.25, 12.5, 25 and 50 µM for 6-OHDA and 0.03, 0.06, 0.125 and 0.25 µM for rotenone. The dataset is balanced, it does not contain any missing values and data were standardized across features. The small number of samples prevented a full and strong statistical analysis of the results. Nevertheless, it allowed the identification of relevant hidden patterns and trends.

Exploratory data analysis, information gain, hierarchical clustering, and supervised predictive modeling were performed using Orange Data Mining version 3.25.1 [42]. Hierarchical clustering was performed using the Euclidean distance metric and weighted linkage. Cluster maps were plotted to relate the features with higher mutual information (in rows) with instances (in columns), with the color of each cell representing the normalized level of a particular feature in a specific instance. The information is grouped both in rows and in columns by a two-way hierarchical clustering method using the Euclidean distances and average linkage. Stratified cross-validation was used to train the supervised decision tree. A set of preliminary empirical experiments were performed to choose the best parameters for each algorithm, and we verified that, within moderate variations, there were no significant changes in the outcome. The following settings were adopted for the decision tree algorithm: minimum number of samples in leaves: 2; minimum number of samples required to split an internal node: 5; stop splitting when majority reaches: 95%; criterion: gain ratio. The performance of the supervised model was assessed using accuracy, precision, recall, F-measure and area under the ROC curve (AUC) metrics.

Regarding classical statistics, the data were analyzed using the Version 8.0.1/2 of GraphPad Software, Inc. (San Diego, CA, USA). The cytotoxicity analysis data are presented as points with a connecting line, Mean ± SEM and statistical relevance was evaluated using Kruskal–Wallis test (one-way ANOVA on ranks) pairwise (control vs. 6-OHDA or control vs. rotenone), (*) *p* < 0.05, (**) *p* < 0.01, (***) *p* < 0.001 and (****) *p* < 0.0001. Seahorse experiment Real-Time ATP Rate Assay data are present in columns and statistical relevance was evaluated using Kruskal–Wallis test (one-way ANOVA on ranks) pairwise (control vs. 6-OHDA or control vs. rotenone) to assess statistical significance of total ATP levels, (*) *p* < 0.05, (**) *p* < 0.01. Additionally, two-way ANOVA with Sidak’s multiple comparisons test was used to assess statistical significance of mitoATP production rate (control vs. 6-OHDA or control vs. rotenone) ($$) *p* < 0.01, ($$$$) *p* < 0.0001) and glycoATP production rate (control vs. 6-OHDA or control vs. rotenone), (###) *p* < 0.001. The remaining results are presented in box-and-whisker plots where the middle line represents the median. The applied method—Min to max—shows all points, plots the whiskers down to the minimal value and up to the maximum value and represents each individual value as a point. Kruskal–Wallis test followed by Dunn’s post hoc analysis was used for multiple conditions comparison. The statistical significance was set at *p* < 0.05 (* *p* < 0.05, ** *p* < 0.01, *** *p* < 0.001 and **** *p* < 0.0001). The number of experiments carried out is represented in the legend of the figures, as well as the number of replicates (at least 2) and the statistical tests applied.

## 5. Conclusions

Together, biological assays and computational methodologies allowed for a distinction between the neurotoxic effects of 6-OHDA and rotenone. For this, features related to mitochondrial physiology, mitochondrial and cellular-derived oxidative stress, and caspase activation, seem to have an enhanced ability to signal neurodegeneration induction, prior to cell death, caused by 6-OHDA and rotenone. This distinction was more evident for the higher concentrations of both compounds that caused more severe and different effects between 6-OHDA and rotenone and between these and control cells. Computational methods also corroborated biological assays in the case of cells treated with lower concentrations of rotenone and 6-OHDA. However, the decision tree algorithm correctly classified 75% of cells treated with 6.25 µM 6-OHDA, which is an improvement concerning the clustering analysis and a finding not foreseen when the biological assays were exclusively taken into consideration.

## Figures and Tables

**Figure 1 ijms-23-03009-f001:**
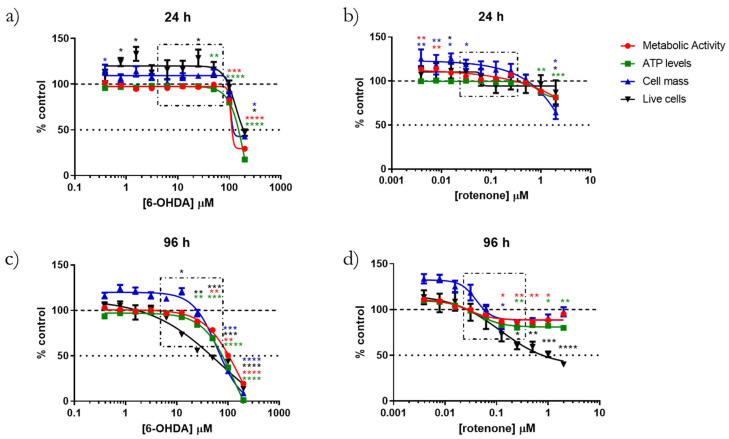
6-OHDA and rotenone induced cytotoxicity in time- and dose-dependent manners. Differentiated SH-SY5Y cells were incubated with increasing concentrations of 6-OHDA or rotenone for 24 h (**a**,**b**, respectively) or for 96 h (**c**,**d**, respectively). Visible alterations in metabolic activity (in red), ATP levels (in green), SRB (in blue) and percentage of live cells (in black) were present in cells treated with both compounds. The compounds concentrations are displayed on the x-axis. Results are presented in percentage of control. Data are represented as Mean ± SEM and four-parameter logistic curves were adjusted to the data. Statistical significance was evaluated using Kruskal–Wallis test (One-way ANOVA on ranks) pairwise (control vs. 6-OHDA or control vs. rotenone), (*) *p* < 0.05, (**) *p* < 0.01, (***) *p* < 0.001 and (****) *p* < 0.0001.

**Figure 2 ijms-23-03009-f002:**
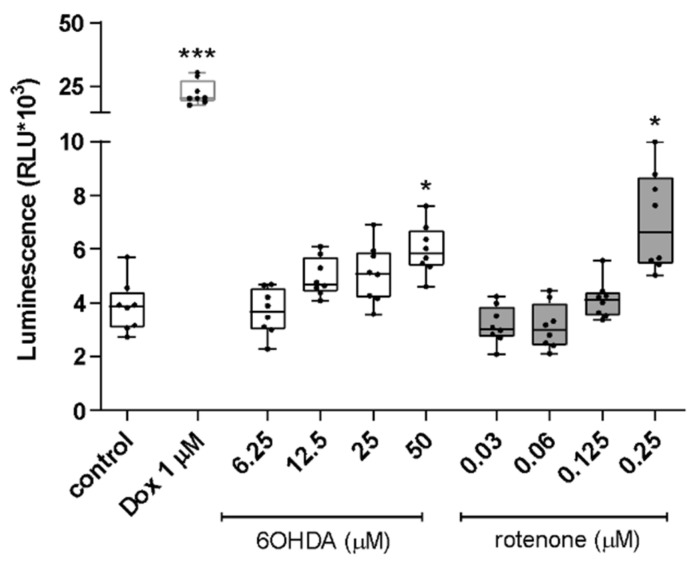
Highest non-lethal concentrations of 6-OHDA (50 µM) and rotenone (0.25 µM) induced an evident increase in caspase 3/7 activity. In order to assess whether non-toxic concentrations of 6-OHDA and rotenone could induce the activation of caspase 3/7, we used the Caspase-Glo 3/7 Assay. Doxorubicin (Dox, 1 µM) was used as a positive control. Data are represented in boxplots in which each dot represents the mean of an independent cell population (*n* = 8), measured in duplicate. Kruskal–Wallis test (One-way ANOVA on ranks) pairwise (control vs. 6-OHDA or control vs. rotenone) was used to assess statistical significance, (*) *p* < 0.05, (***) *p* < 0.001.

**Figure 3 ijms-23-03009-f003:**
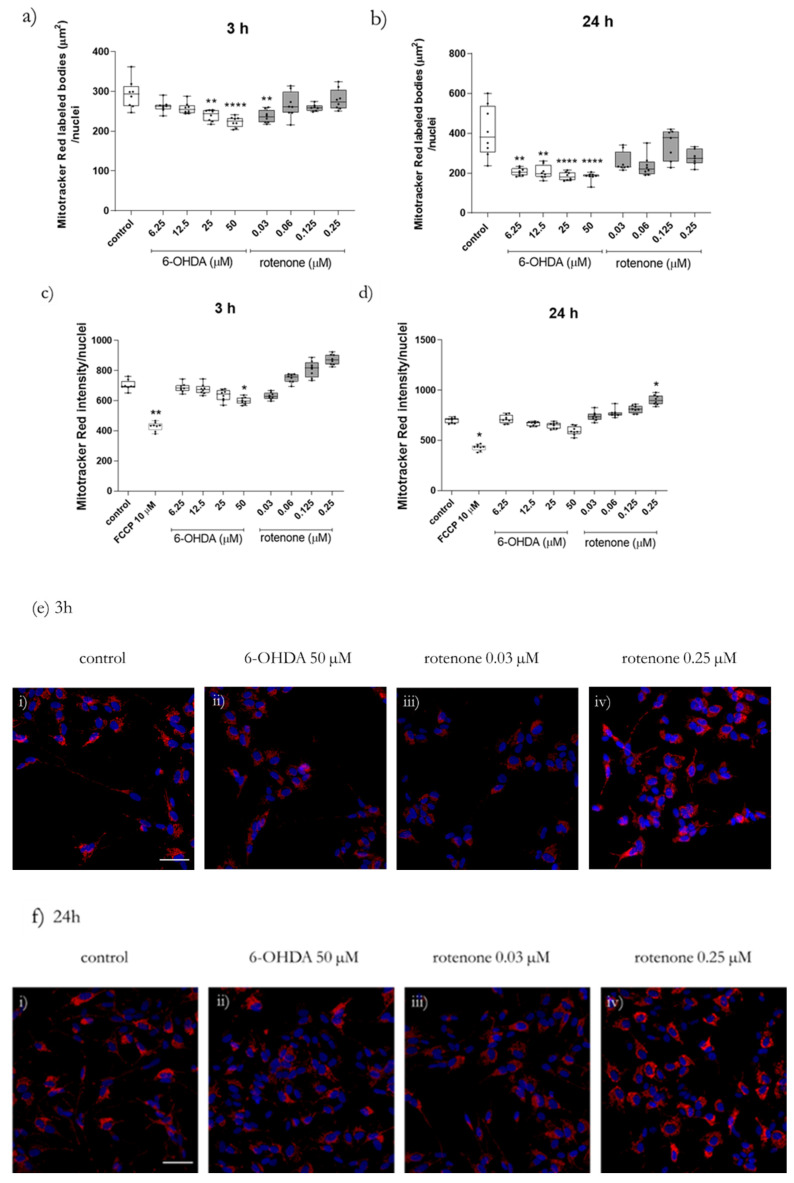
Non-toxic concentrations of 6-OHDA and rotenone prompted alterations in Mitotracker area and fluorescence intensity. Mitotracker Red labeled bodies area and intensity were evaluated in cells treated with 6-OHDA and rotenone for 3 h (**a**,**c**) and 24 h (**b**,**d**) with the mitochondrial membrane potential-dependent dye Mitotracker Red CMXRos. Data are represented in boxplots in which each dot represents an independent cell population (*n* = 8), in duplicate. Kruskal–Wallis test (one-way ANOVA on ranks) pairwise (control vs. 6-OHDA or control vs. rotenone) was used to assess statistical significance, (*) *p* < 0.05, (**) *p* < 0.01, (****) *p* < 0.0001. Live-cell microscopy images of Mitotracker Red CMXRos labeling in SH-SY5Y cells were acquired using a 20× objective in IN Cell Analyzer 2200. Scale bar = 50 µm. Nuclei staining (Hoechst 33,342) is presented in blue, and Mitotracker Red CMXRos is presented in red (**e**,**f**). Representative images of cells treated for 3 h (**c**) or 24 h (**d**) with 50 µM 6-OHDA (**ii**) or with 0.03 (**iii**) or 0.25 µM (**iv**) rotenone are shown. Non-treated cells are presented in (**i**) in both (**e**,**f**) panels.

**Figure 4 ijms-23-03009-f004:**
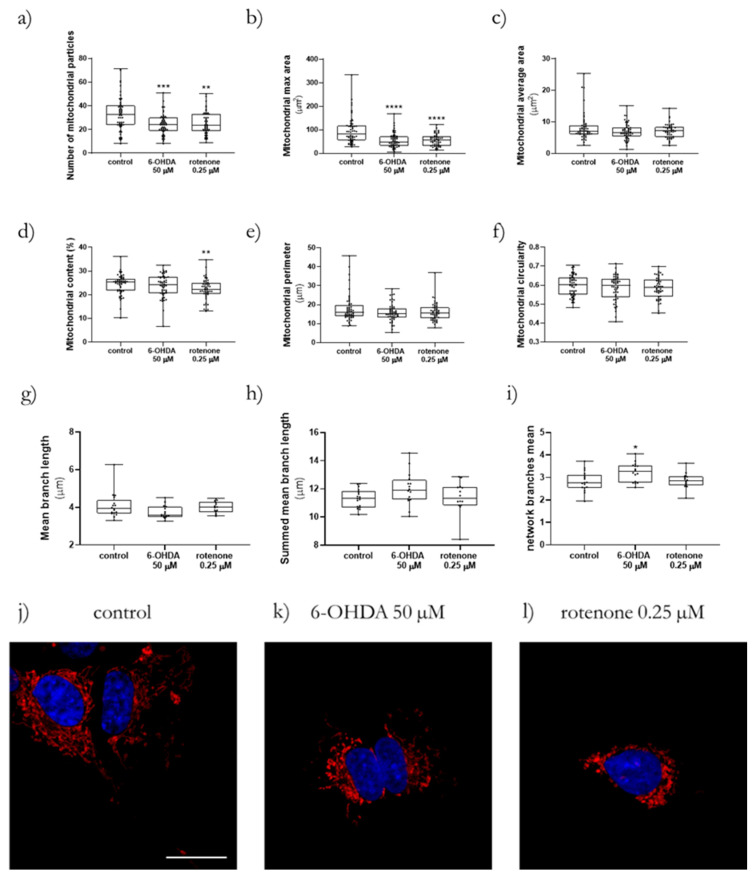
Non-toxic concentrations of 6-OHDA and rotenone promoted alterations in mitochondrial morphology. The mitochondrial morphology parameters number of mitochondrial particles (**a**), maximal mitochondrial area (**b**), mitochondrial average area (**c**), mitochondrial content (**d**), mitochondrial perimeter (**e**), and mitochondrial circularity (**f**) were analyzed using the Mito-Morphology macro, while the mitochondrial mean branch length (**g**), summed mean branch length (**h**), and network branches mean (**i**) were evaluated using the MiNA toolset. Mito-Morphology data are represented in boxplots in which each dot represents an individual cell (*n* = 49–55). MiNA toolset data are represented in boxplots in which each dot represents an independent cell population (*n* = 14–17). Kruskal–Wallis test (one-way ANOVA on ranks) pairwise (control vs. 6-OHDA or control vs. rotenone) was used to assess statistical significance, (*) *p* < 0.05, (**) *p* < 0.01, (***) *p* < 0.001 and (****) *p* < 0.0001. Representative images were obtained by confocal microscopy (63× objective), scale bar = 20 µm. Nuclei staining is presented in blue and Mitotracker Red CMXRos is presented in red (**j**–**l**). Cells were treated 24 h with 50 µM 6-OHDA (**k**) or with 0.25 µM rotenone (**l**). Non-treated cells are presented in (**j**).

**Figure 5 ijms-23-03009-f005:**
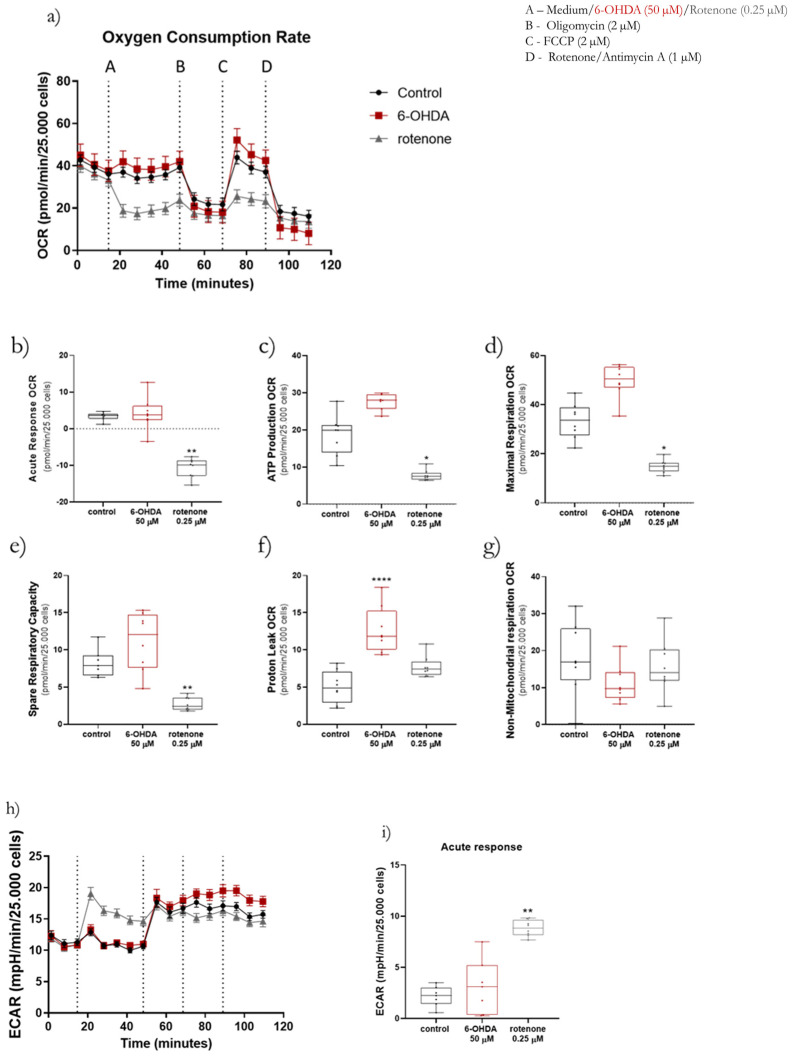
Acute treatments with 50 µM 6-OHDA and with 0.25 µM rotenone differently affected mitochondria bioenergetics. Differentiated SH-SY5Y were acutely treated with 50 µM 6-OHDA or with 0.25 µM rotenone just prior to the conventional Seahorse XF Cell Mito Stress Test. OCR was assessed over time with the addition of 6-OHDA or rotenone (A), Oligomycin (B), FCCP (C), and rotenone/antimycin A (D) (**a**). Mitochondrial bioenergetic parameters acute response (**b**), ATP production (**c**), maximal respiration (**d**), spare respiratory capacity (**e**), proton-leak (**f**) and non-mitochondrial respiration (**g**) -linked OCR, as well as the ECAR profile (**h**,**i**) were distinctly affected by 6-OHDA and rotenone. Data are represented in boxplots in which each dot represents the mean of an independent cell population (*n* = 8). Kruskal–Wallis test (one-way ANOVA on ranks) pairwise (control vs. 6-OHDA or control vs rotenone) was used to assess statistical significance, (*) *p* < 0.05, (**) *p* < 0.01, (****) *p* < 0.0001.

**Figure 6 ijms-23-03009-f006:**
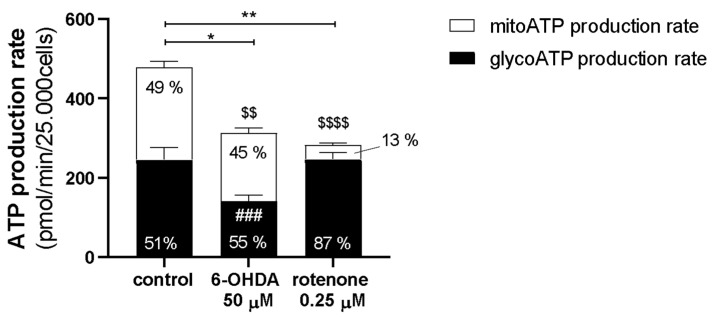
Acute treatments with 50 µM 6-OHDA and with 0.25 µM rotenone decreased total ATP levels in distinct manners. Cells acutely incubated with 50 µM 6-OHDA and with 0.25 µM rotenone exhibited a decrease in total ATP levels. Black bars represent glycoATP production rate, while white bars represent mitoATP production rate. Data are presented in columns representing the Mean ± SEM of 8 independent cell populations. Kruskal–Wallis test (one-way ANOVA on ranks) pairwise (control vs. 6-OHDA or control vs. rotenone) was used to assess statistical significance of total ATP levels, (*) *p* < 0.05, (**) *p* < 0.01. Additionally, two-way ANOVA with Sidak’s multiple comparisons test was used to assess statistical significance of mitoATP production rate (control vs. 6-OHDA or control vs rotenone) ($$) *p* < 0.01, ($$$$) *p* < 0.0001, and glycoATP production rate (control vs. 6-OHDA or control vs. rotenone), (###) *p* < 0.001.

**Figure 7 ijms-23-03009-f007:**
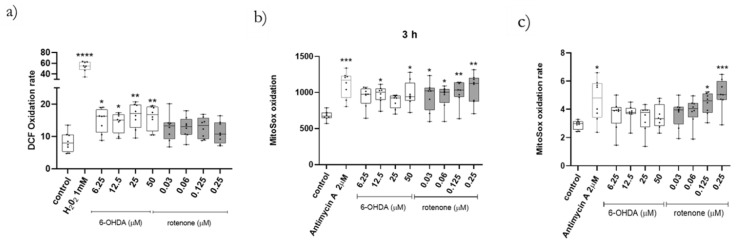
Treatments with both compounds induced an increase in cellular and mitochondrial-derived oxidative stress The fluorogenic substrates DCF (**a**) and Mitosox (**b**,**c**) were used to study, respectively, cellular and mitochondrial-derived oxidative stress in cells treated with different concentrations of 6-OHDA and rotenone. Cells were either incubated acutely (**a**,**c**) or for 3 h (**b**) with different concentrations of either compound. Antimycin A, 2 µM, was used as positive control. Data are represented in boxplots in which each dot represents an independent cell population (*n* = 8). Kruskal–Wallis test (one-way ANOVA on ranks) pairwise (control vs. 6-OHDA or control vs. rotenone) was used to assess statistical significance, (*) *p* < 0.05, (**) *p* < 0.01, (***) *p* < 0.001 and (****) *p* < 0.0001.

**Figure 8 ijms-23-03009-f008:**
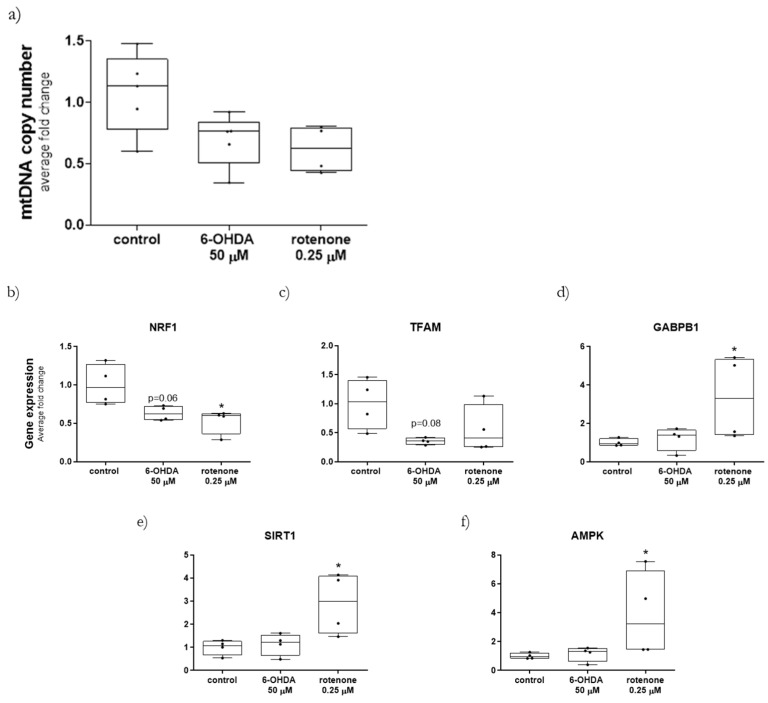
mRNA levels of mitochondrial biogenesis-associated genes and mtDNA copy number were differently affected by 6-OHDA and rotenone. Mitochondria DNA copy number (**a**) and mitochondrial biogenesis-associated genes NRF1 (**b**), TFAM (**c**), GABPB1 (**d**), SIRT1 (**e**) and AMPK (**f**) were measured in cells treated with 50 µM 6-OHDA and with 0.25 µM rotenone for 24 h. Data are represented in boxplots in which each dot represents an independent cell population (*n* = 4). Kruskal–Wallis test (one-way ANOVA on ranks) pairwise (control vs. 6-OHDA or control vs. rotenone) was used to assess statistical significance, (*) *p* < 0.05.

**Figure 9 ijms-23-03009-f009:**
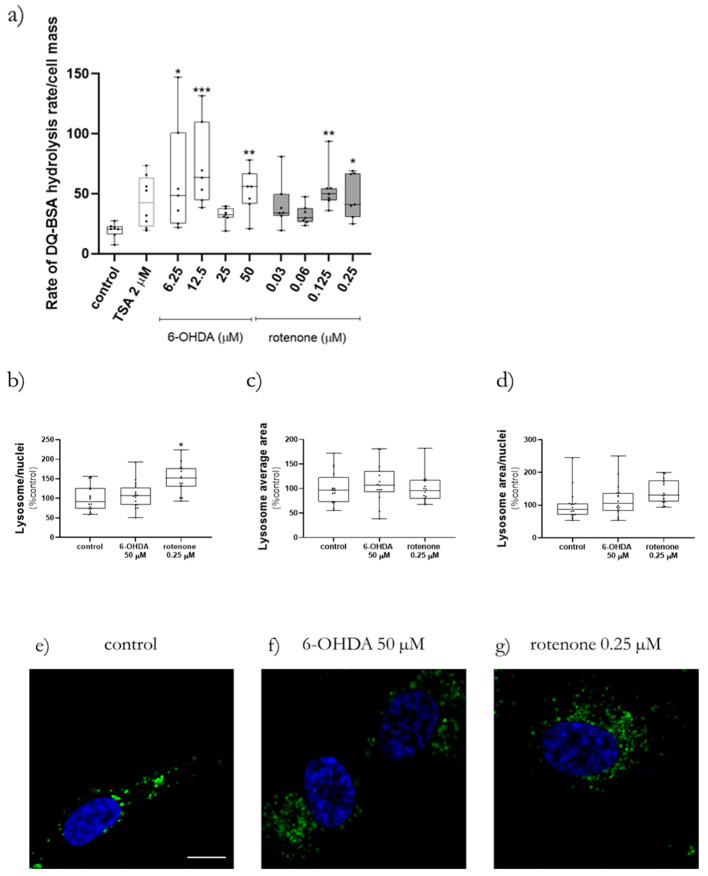
6-OHDA and rotenone promoted the activation of lysosomal protease activity as well as the increase in the number of lysosomes per cell. The fluorogenic substrate DQ Green BSA was used to assess lysosomal protease activity in cells acutely treated with different concentrations of 6-OHDA and rotenone (**a**). Thrichostatin A (TSA), 2 µM, was used as positive control. Data are represented in boxplots in which each dot represents the average of an independent cell population (*n* = 7). Lysosomal parameters lysosome/nuclei (**b**), lysosome average area (**c**) and lysosome area/nuclei (**d**) were also analyzed and are presented in percentage of control. Data are represented in boxplots in which each dot represents the average of an independent cell population (*n* = 15–16). Kruskal-Wallis test (One-way ANOVA on ranks) pairwise (control vs. 6-OHDA or control vs. rotenone) was used to assess statistical significance, (*) *p* < 0.05, (**) *p* < 0.01, (***) *p* < 0.001. Representative images were obtained by confocal microscopy (63x objective), scale bar = 10 µm. Nuclei staining is presented in blue and lysotracker green is presented in green (**e**–**g**). Cells were treated for 24 h with 50 µM 6-OHDA (**f**) or with 0.25 µM rotenone (**g**). Non-treated cells are presented in (**e**).

**Figure 10 ijms-23-03009-f010:**
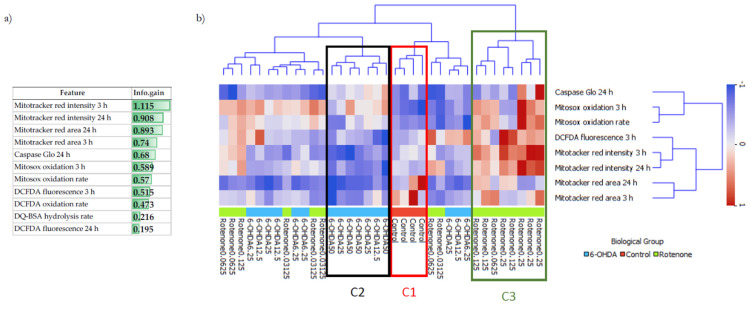
Hierarchical clustering, with a subset of features presenting an information gain higher than 0.5, could distinctly group control cells and cells treated with different concentrations of 6-OHDA or rotenone. Information gain analysis of the 11 features measured in this work (**a**). The tree’s leaves resulting from hierarchical clustering were labeled using both biological groups and the corresponding concentrations. The subset of features used for clustering are presented in the rows of the heatmap (**b**).

**Table 1 ijms-23-03009-t001:** Decision tree confusion matrix. The test metrics of the decision tree are the following: {Accuracy: 0.5; Precision: 0.52; Recall; 0.5; F1-score: 0.48; AUC: 0.74}. Values on the confusion matrix correspond to test results and are presented as the percentage of the proportion of the actual class.

		Predicted Label								
		6-OHDA 50 µM	6-OHDA 25 µM	6-OHDA 12.5 µM	6-OHDA 6.25 µM	Control	Rotenone 0.03 µM	Rotenone 0.06 µM	Rotenone 0.125 µM	Rotenone 0.25 µM
True label	6-OHDA 50 µM	75	25	0	0	0	0	0	0	0
	6-OHDA 25 µM	50	25	25	0	0	0	0	0	0
	6-OHDA 12.5 µM	0	25	25	50	0	0	0	0	0
	6-OHDA 6.25 µM	0	0	0	75	0	0	0	25	0
	Control	0	0	0	25	50	0	0	0	25
	Rotenone 0.03 µM	0	0	0	25	0	25	25	25	0
	Rotenone 0.06 µM	0	0	0	25	0	25	25	25	0
	Rotenone 0.125 µM	0	0	0	0	0	0	25	50	25
	Rotenone 0.25 µM	0	0	0	0	0	0	0	0	100

**Table 2 ijms-23-03009-t002:** List of forward (Fwd) and reverse (Rev) primers used for real time-PCR, respective gene, accession number, and annealing temperature (Ta) for genes used in the calculation of mtDNA copy number and for genes encoding electron transport chain complex and ATP synthase subunits, proteins involved in mitochondrial fission and fusion, antioxidant enzymes, mitochondrial biogenesis-related gene transcripts, and mitochondrial biogenesis-related gene transcripts. *CYB*: cytochrome b; *B2M*: β2 microglobulin; *ATP5G1*: ATP synthase membrane subunit c locus 1; *ATP6*: ATP synthase membrane subunit 6; *COX1*: cytochrome c oxidase I; *COX4I1*: cytochrome c oxidase subunit 4I1; CYB: cytochrome b; ND5: NADH: Ubiquinone oxidoreductase core subunit 5; *NDUFA9*: NADH: Ubiquinone oxidoreductase subunit A9; *SDHA*: succinate dehydrogenase complex flavoprotein subunit A; *UQCRC2*: ubiquinol-cytochrome c reductase core protein 2; *DRP1*: dynamin-related protein 1; *FIS1*: mitochondrial fission 1 protein; *MFF*: mitochondrial fission factor; *MFN1*: mitofusin 1; *MFN2*: mitofusin 2; *MIEF2*: mitochondrial elongation factor 2; *OPA1*: optic atrophy 1 protein; *CAT*: catalase; *GPX1*: glutathione peroxidase 1; *GPX4*: glutathione peroxidase 4; *SOD1*: superoxide dismutase 1; *SOD2*: superoxide dismutase 2; *AMPK*: adenosine monophosphate-activated protein kinase; *GABPB1*: GA binding protein transcription factor subunit beta 1; *NRF1*: nuclear respiratory factor 1; *SIRT1*: sirtuin 1; *TFAM*: mitochondrial transcription factor A.

Gene	Accession Number	Fwd Primer	Rev Primer	Ta (°C)
*AMPK*	NM_006251	TCCGTAGTATTGATGATGAAAT	TTAGGTCAACAGGAGAAGAG	60
*ATP5G1*	NM_001002027	GGCTAAAGCTGGGAGACTGAAA	GTGGGAAGTTGCTGTAGGAAGG	60
*ATP6*	NC_012920 (8527-9207)	GCGCCACCCTAGCAATATCA	GCTTGGATTAAGGCGACAGC	60
*B2M*	NC_000015	TGTTCCTGCTGGGTAGCTCT	CCTCCATGATGCTGCTTACA	60
*CAT*	NM_001752.4	CTCAGGTGCGGGCATTCTAT	TCAGTGAAGTTCTTGACCG	60
*COX1*	NC_012920 (5904–7445)	ATACCAAACGCCCCTCTTCG	TGTTGAGGTTGCGGTCTGTT	60
*COX4I1*	NM_001861	GAGAAAGTCGAGTTGTATCGCA	GCTTCTGCCACATGATAACGA	60
*CYB*	NC_012920 (14747–15887)	CCACCCCATCCAACATCTCC	GCGTCTGGTGAGTAGTGCAT	60
*CYB*	NC_012920 (14747–15887)	CCACCCCATCCAACATCTCC	GCGTCTGGTGAGTAGTGCAT	60
*DRP1*	NM_012063	TCCAGCTGCCTCAAATCGTC	TGCTTCCACCCCATTTTCTTCT	60
*FIS1*	NM_016068	AGCGGGATTACGTCTTCTACC	CATGCCCACGAGTCCATCTTT	60
*GABPB1*	NM_005254	GCCACAGAAGAAGTAGTTAC	ACTGTTAATACTTGTTGTCCAT	60
*GPX1*	NM_000581.4	GGAGAACGCCAAGAACGAA	TTCTCGAAGAGCATGAAGT	60
*GPX4*	NM_002085.5	AAGATCTGCGTGAACGGGG	CCACTTGATGGCATTTCCCAG	60
*MFF*	NM_001277061.2	CTTGGATGTGCTGGATGA	TTTGATTATCTGTCGTCTTAGTGA	60
*MFN1*	NM_033540	AGTTGGAGCGGAGACTTAGCA	TTCTACCAGATCATCTTCAGTGGC	60
*MFN2*	NM_014874	AGCTACACTGGCTCCAACTG	AACCGGCTTTATTCCTGAGCA	60
*MIEF2*	NM_139162.4	CAATCCACCAACAGAATG	CGTTAAGTCACCTTCTCT	60
*ND5*	NC_012920 (12337–14148)	AGTTACAATCGGCATCAACCAA	CCCGGAGCACATAAATAGTATGG	60
*NDUFA9*	NM_005002	GCCTATCGATGGGTAGCAAGAG	TGAGTTCCAGTGGTGTTGCC	60
*NRF1*	NM_005011	TTGAGTCTAATCCATCTATCCG	TACTTACGCACCACATTCTC	60
*OPA1*	NM_015560	GCTCTAAACCATTGTAACCTTTGT	TTCTCTAATCGCCTAACTTCAGT	60
*SDHA*	NM_004168	CGGGTCCATCCATCGCATAAG	TATATGCCTGTAGGGTGGAACTGAA	60
*SIRT1*	NM_012238	GTAGGCGGCTTGATGGTAAT	GGGTTCTTCTAAACTTGGACTCT	60
*SOD1*	NM_000454	CGAGCAGAAGGAAAGTAATG	GGATAGAGGATTAAAGTGAGGA	60
*SOD2*	NM_000636	GAAGTTCAATGGTGGTGGTCAT	TTCCAGCAACTCCCCTTTGG	60
*TFAM*	NM_003201	GTTTCTCCGAAGCATGTG	GGTAAATACACAAAACTGAAGG	60
*UQCRC2*	NM_003366	TTCAGCAATTTAGGAACCACCC	GGTCACACTTAATTTGCCACCAA	60

## Data Availability

The dataset generated during and/or analyzed during the current study are available on https://doi.org/10.6084/m9.figshare.19146410.

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
