# Peer review of "Evaluation of 6-Hydroxydopamine and Rotenone In Vitro Neurotoxicity on Differentiated SH-SY5Y Cells Using Applied Computational Statistics"

_ijms, 2022, doi:10.3390/ijms23063009_

Round 1
Reviewer 1 Report
Simões et al evaluated the differentiated SH-SY5Y cells treated with 6-hydroxydopamine (6-OHDA) and rotenone concerning cell viability, mitochondrial morphology, physiology, associated gene expression, ATP production, the lysosome. Combined with machine learning methods, the authors aimed to determine the experimental biological features most relevant in neurodegeneration induction before cell death. The results suggest that mitochondrial physiology, mitochondrial and cellular derived oxidative stress, and Caspase activity are relevant in neurodegeneration induction, prior to cell death. The decision tree algorithm helps interpret the data and find the best concentration for evaluation.
The neurotoxicities of 6-OHDA and rotenone (Mitochondrial complex I inhibitor) have been known to correlate with mitochondria, autophagy-lysosomal pathways. Simões et al’s study complements the previous findings with more detailed evidence like seahorse data, morphological observation, and gene analyses. The data quality looks good. The supplementary material is not available to review. Furthermore, the unsupervised and supervised machine learning methods in this study introduce a new way to analyze the data.
Questions: The current study still focuses on the known pathways like mitochondria, oxidative stress, and autophagy-lysosomal pathways. The aim of this work is to determine the features relevant in neurodegeneration, however, other possible pathways of 6-OHDA like neuroinflammation (PMID: 11918659, PMID: 18545225) are not involved or discussed. Have the authors already excluded these pathways in neurodegeneration?
Author Response
Point 1: Simões et al evaluated the differentiated SH-SY5Y cells treated with 6-hydroxydopamine (6-OHDA) and rotenone concerning cell viability, mitochondrial morphology, physiology, associated gene expression, ATP production, the lysosome. Combined with machine learning methods, the authors aimed to determine the experimental biological features most relevant in neurodegeneration induction before cell death. The results suggest that mitochondrial physiology, mitochondrial and cellular derived oxidative stress, and Caspase activity are relevant in neurodegeneration induction, prior to cell death. The decision tree algorithm helps interpret the data and find the best concentration for evaluation.
The neurotoxicities of 6-OHDA and rotenone (Mitochondrial complex I inhibitor) have been known to correlate with mitochondria, autophagy-lysosomal pathways. Simões et al’s study complements the previous findings with more detailed evidence like seahorse data, morphological observation, and gene analyses. The data quality looks good. The supplementary material is not available to review. Furthermore, the unsupervised and supervised machine learning methods in this study introduce a new way to analyze the data.
Questions: The current study still focuses on the known pathways like mitochondria, oxidative stress, and autophagy-lysosomal pathways. The aim of this work is to determine the features relevant in neurodegeneration, however, other possible pathways of 6-OHDA like neuroinflammation (PMID: 11918659, PMID: 18545225) are not involved or discussed. Have the authors already excluded these pathways in neurodegeneration?
Response 1 : This is a very pertinent comment. As described in the introductory section (lines 39-44), we acknowledge that neuroinflammation (as well as other factors) could have an important role in neurodegeneration. It was previously shown that both rotenone and 6-OHDA can induce neuroinflammation in different models. An increase in proinflammatory cytokines, both in vitro and in vivo, and recruitment of immune cells to different brain areas were previously described. However, in this manuscript, we focused on the effect of these compounds in cell and mitochondrial metabolism, oxidative stress and autophagosomal-lysosomal pathways.
Regarding the supplementary material, we now provide it in the revised version of the manuscript
Reviewer 2 Report
The manuscript by Simoes et al employs computational analysis of the effects of 6-OHDA and rotenone - two neurotoxins widely used as models of Parkinson's Disease – on an array of biochemical parameters measured in SH-SY5Y cells. While individual data pieces are not novel, the analytical approach is interesting and yields novel perspective on the interpretation of the results. The manuscript is clearly written and easy to follow. I do, however, have several comments that the authors need to address before the manuscript is ready for publication.
Major:
- I am somewhat confused by the aims of this study. The authors state that they want to investigate homeostatic changes in toxin-treated cells before the onset of cell death. Why then the treatments range from acute to 3h and 24h exposure? It's likely that only the longest exposure times with the toxins may be related to PD-like alteration in cellular biology and metabolism. On the other hand, study's current design does address differences between toxicity mechanisms of 6-OHDA and rotenone (i.e. regardless of their relevance to PD), but for the clarity of presentation the authors need to make sure the difference between acute exposure to toxins and compensatory effects at longer exposures are emphasized and discussed.
- For some reason, only a portion of the experimental data are used for clustering analysis. Why, for example, Seahorse, mtDNAcn and qPCR data are left out? Please, provide inclusion or exclusion criteria and justifications. More importantly, if the experimental data are not used for the follow up clustering analysis, I recommend not showing or moving these data to supplementary.
- Fig 11 analysis is only based on 4 data sets for each group, so the statistical accuracy of the predictions is somewhat limited.
Minor:
- Fig 3: the last two panels are mislabeled.
- There is a contradiction between Lines 391-392 that says that DQ Green BSA hydrolysis rate was measured after acute treatment 6-OHDA and rotenone, and Fig 9 legend that claims that cells were treated with the toxins for 24h.
Author Response
The manuscript by Simoes et al employs computational analysis of the effects of 6-OHDA and rotenone - two neurotoxins widely used as models of Parkinson's Disease – on an array of biochemical parameters measured in SH-SY5Y cells. While individual data pieces are not novel, the analytical approach is interesting and yields a novel perspective on the interpretation of the results. The manuscript is clearly written and easy to follow. I do, however, have several comments that the authors need to address before the manuscript is ready for publication.
Major:
Point 1: I am somewhat confused by the aims of this study. The authors state that they want to investigate homeostatic changes in toxin-treated cells before the onset of cell death. Why then the treatments range from acute to 3h and 24h exposure? It's likely that only the longest exposure times with the toxins may be related to PD-like alteration in cellular biology and metabolism. On the other hand, study's current design does address differences between toxicity mechanisms of 6-OHDA and rotenone (i.e. regardless of their relevance to PD), but for the clarity of presentation the authors need to make sure the difference between acute exposure to toxins and compensatory effects at longer exposures are emphasized and discussed.
Response 1: Thank you for the comment. In general, we started our experiments by treating cells with different concentrations of either compound for 24h. Our experimental design considered the time course of different events in response to mitochondrial damage. If effects on mitochondria are due to direct interaction with the compounds, they should be almost immediate. In turn, caspase activation, for example, should be a later consequence. This needs to be taken into account when deciding each timepoints for each parameter. This way, our intent with shorter timepoints (acute and 3 h treatments) was aimed to understand if and when the toxic effects of rotenone and 6-OHDA would occur.
Point 2: For some reason, only a portion of the experimental data are used for clustering analysis. Why, for example, Seahorse, mtDNAcn and qPCR data are left out? Please, provide inclusion or exclusion criteria and justifications. More importantly, if the experimental data are not used for the follow up clustering analysis, I recommend not showing or moving these data to supplementary.
Response1: We appreciate the comment made by the reviewer. For the clustering analysis, only the experiments in which the cells were treated with 6.25, 12.5, 25 and 50 µM for 6-OHDA and 0.03, 0.06, 0.125 and 0.25 µM for rotenone were taken into consideration. This allowed us to discriminate, not only between the different compounds, but also between the different concentrations of both compounds, allowing us to ascertain the effects of different degrees of neurotoxicity. It is expected that increasing concentrations of mitochondrial toxins (6-OHDA and rotenone) will have gradually more evident effects, reaching a phase in which the neurotoxicity threshold is crossed (point-of-no-return). This explains why it is more interesting to study machine-learning with these parameters. The parameters measured with selected concentrations serve to characterize the level of mito- and neurotoxicity of each agent.
Moreover, we wanted to show that our computational models were able to distinguish between treated and untreated (control) cells, even with smaller concentrations of 6-OHDA and rotenone, shown to induce an effect more similar to untreated cells. This is clearly evident in the cluster analysis. As stated in the manuscript, section 2.8, “Hierarchical clustering perfectly separated control cells from cells treated with different concentrations of both treatments (...) we could isolate control cells (red box, C1). A different cluster (Black box, C2) grouped cells treated with higher 6-OHDA concentrations (25 µM and 50 µM). Cluster C3 (green box) contained cells treated with different rotenone concentrations. The remaining clusters grouped cells treated either with 6-OHDA or rotenone. Interestingly, these clusters contained cells treated with lower concentrations of both compounds, shown to induce small and similar effects.” Additionally, our decision tree model was able to correctly predict 75% of cells treated with 6.25 µM 6-OHDA, being an improvement from the results derived from the hierarchical clustering analysis (as stated in section 2.9).
The remaining experiments, in which the protocols were only performed using control cells and cells treated for 24 h with 50 μM 6-OHDA and 0.25 μM rotenone were not taken into consideration in the computational analysis since the highest concentrations of both compounds were the ones displaying more drastic effects, when compared to control cells, and so perfect clustering separation was to be expected. Moreover, since these experiments did not have the 36 samples of the other experiments (but only 12 samples), they could not be included in the analysis.
Regarding not showing the data or moving it to supplementary material, we think that despite not being included into the computational analysis, this data is still of key importance to understand the mechanism of action of both 6-OHDA and rotenone, since it is important to characterize the level of mito- and neurotoxicity of each agent.
Point 3: Fig 11 analysis is only based on 4 data sets for each group, so the statistical accuracy of the predictions is somewhat limited.
Response 3: Thank you for this comment. However, as stated in section 4.16 "The complete dataset used in the analysis comprises 36 samples, each described by 11 numeric features and 1 target. (...) The small number of samples prevented a full and strong statistical analysis of the results. Nevertheless, it allowed the identification of relevant hidden patterns and trends.” The differences observed between cells of the same class represent experimental variability (and not biological variability) since cells from each class have the same biological origin (relatively homogenous system).
Minor:
Point 1: Fig 3: the last two panels are mislabeled.
Response 1: We accept and appreciate the suggestion. We have corrected this in the new manuscript version.
Point 2: There is a contradiction between Lines 391-392 that says that DQ Green BSA hydrolysis rate was measured after acute treatment 6-OHDA and rotenone, and Fig 9 legend that claims that cells were treated with the toxins for 24h.
Response 2: This is a very pertinent comment. We have corrected this in the new manuscript version. In the figure 9 caption is now written “The fluorogenic substrate DQ Green BSA was used to assess lysosomal protease activity in cells acutely treated with different concentrations of 6-OHDA and rotenone (a)”
Round 2
Reviewer 2 Report
While I appreciate author's detailed answers to my questions, I disagree with their hesitance to restructure the manuscript as I've suggested.
The main point of our disagreement, as I see it, is whether data from acutely treated cells and those exposed to the toxins for 3h and 24h should be analyzed together or separately in clustering analysis. As we both agree, acute exposure to toxins and compensatory effects at longer exposures represent different cellular processes, but in the current manuscript all 3 conditions are treated as equal predictors of the outcome of toxin exposure. In my opinion, this diminishes the value of presented analytical approach as it is quite rare that exposure to a toxic insult is analyzed at multiple timepoints. A more common scenario is that data are available from either acute or long exposure and the important question that the authors can easily address in this manuscript is "how well clustering analysis can separate treatments groups at different exposure time points?". Are control and treated cells more dissimilar after acute exposure and become less segregated later or the other way around? Such analysis would be both more meaningful and more helpful as it can be used for future analysis of various animal and cell disease models.
Author Response
A: These time points were used because they translate the early cellular response to mitochondrial toxicity. However, we agree with the reviewer that this point was not sufficiently clear, and thus we introduced new information and terminology in the manuscript.
The main goal of this work was to study early mitochondrial changes induced by two well-known mitochondrial toxins (6-hydroxydopamine and rotenone) that trigger later neurodegeneration and cell death. Thus, the first task was to select a range of concentrations of each toxin that do not induce detectable changes in cell viability until 24 h, but which lead to cell death after longer incubation times (96 h), as described in section 2.1 of the manuscript. We can, thus, consider all the measurements taken before or at 24 h as being early predictors of later neurotoxic manifestations. As we mentioned in the previous revision, it is essential to choose the appropriate timepoints to measure each biochemical parameter, taking into account the time course of the cellular response. As the selected toxins are expected to interact directly with mitochondria, acute measurements of mitochondrial function, shortly after exposure, should be highly informative. However, the cellular response to mitochondrial dysfunction can further affect mitochondrial function, either by compensatory changes or by the commitment to cell death pathways. By combining information obtained before neuronal cell death, and at different time points according to the nature of cellular signaling cascades, we can clearly identify the early response to mitochondrial toxicity that culminates in neurodegeneration.
The features selected for computational data analysis cover the immediate response (acute exposure) and two time points that can inform on the success of compensatory responses (3 and 24 h).
New information related to this question can now be found in section 2.1, page 3, line 106 and 107, and section 2.8, page 18, lines 430 to 431.